# New insights into the spatial organization, stratigraphy and human occupations of the Aceramic Neolithic at Ganj Dareh, Iran

**Julien Riel-Salvatore**[1]*, **Andrew Lythe**[1], **Alejandra Uribe Albornoz**[2]

**1** Département d'Anthropologie, Laboratoire d'Archéologie de l'Anthropocène, Université de Montréal, Montréal, QC, Canada, **2** Département de Géographie, Université de Montréal, Montréal, QC, Canada

* julien.riel-salvatore@umontreal.ca

**Data Availability Statement:** All relevant data are within the manuscript and its S1 File and S1 Video.

**Funding:** This work was funded and supported by Université de Montréal, and the Canadian

## Abstract

The Aceramic Neolithic site of Ganj Dareh (Kermanshah, Iran) is arguably one of the most significant sites for enhancing our understanding of goat domestication and the onset of sedentism. Despite its central importance, it has proven difficult to obtain contextually reliable data from it and integrate the site in regional syntheses because it was never published in full after excavations ceased in 1974. This paper presents the Ganj Dareh archive at Université de Montréal and shows how the documentation and artifacts it comprises still offer a great deal of useful information about the site. In particular, we 1) present the first stratigraphic profile for the site, which reveals a more complex depositional history than Smith's five-level sequence; 2) reveal the presence of two possible pre-agricultural levels (H-01 and P-01); 3) explore the spatial organization of different levels; 4) explain possible discrepancies in the radiocarbon dates from the site; 5) show some differences in lithic technological organization in levels H-01 and P-01 suggestive of higher degrees of residential mobility than subsequent phases of occupation at the site; and 6) reanalyze the burial data to broaden our understanding of Aceramic Neolithic mortuary practices in the Zagros. These data help refine our understanding of Ganj Dareh's depositional and occupational history and recenter it as a key site to improve our understanding the Neolithization process in the Middle East.

## Introduction

Western Iran, in particular the central Zagros and its foothills, was from the 1950s to the 1970s, a hotbed of research into the shifts from hunting and gathering to food production [1–7]. Much of this research was focused specifically on the 'economic' aspects of the Neolithization process, in particular as it concerned the process of animal domestication [8–10]. Changing political conditions largely brought an end to this early phase of research, which would resume at the turn of the 21st Century with new field projects (e.g., [11–14]) and reexcavations of several major sites identified during the first wave of research [15, 16]. Importantly, new research spearheaded by Iranian archaeologists is bringing to light important new data on

Foundation for Innovation John R. Edward Leaders Fund grant #37754 (to JRS). AUA is supported by a Joseph-Armand Bombardier Canada Graduate Scholarship.

**Competing interests:** The authors have declared that no competing interests exist.

areas that had been explored using different theoretical and methodological frameworks (e.g. [14, 17]), highlighting the social dimensions of the Neolithization process (e.g., [13]).

From this perspective, modern reanalyses of material from sites excavated over a generation ago are yielding new information about aspects linked to mortuary practices and social identity (e.g., [18]). For instance, recent bioarchaeological and genetic work on the human remains from the site of Tepe Abdul Hosein, first excavated in 1978 [19], have shown the presence of underappreciated variability in mortuary practices, including extreme cultural cranial modification and of the indigenous development of the Neolithic in the Eastern Fertile Crescent [20, 21]. However, it has not been possible to integrate material from some ancient 'key' sites, especially those that have never been the subject of a synthetic monographic publication. One such case is the site of Ganj Dareh Tappeh.

Ganj Dareh Tappeh, located in the Kermanshah province of Iran, is an Aceramic Neolithic site that has yielded the earliest evidence of goat domestication in the world and appears to have been intensely occupied for a 200–300 year span around 10,000 cal $^{14}$C yrs ago. Recent paleo-environmental work in the Central Zagros indicates that this was a period of dry summers and wet winters, with a markedly seasonal precipitation regime; it was associated with the establishment of *Pistacia* by ca. 10,000 cal BP, with *Quercus* beginning to spread across the region around that time [22, 23]. This regime replaced the herb steppe that had characterized the region and the rest of the Iranian Plateau during the Younger Dryas and made it comparatively poor in resources and perhaps depopulated by humans [24]. As such, its chronology indicates that Ganj Dareh has the potential to provide a unique glimpse into the behavioral dynamics that accompanied this fundamental shift in our species' evolution. However, gleaning this information has to date proved impossible because the site itself has only been minimally published. Notably, while the original excavator describes it as comprising a series of five distinct levels containing different records and feature types [25, 26], the characteristics of each level have never been published in detail and little data on the various technologies used by humans has been made available.

To fill in some of these gaps, this paper presents an in-depth description of the different sedimentary units that can be identified at the site and a first reconstruction of its stratigraphy. These are accompanied by a description of features and architecture in each of the 14 identified stratigraphic units, which include firepits, mud-brick architecture, hearths and burials. In addition, a preliminary analysis of the changing densities of lithic and ceramic artifacts is presented in order to address the question of whether the adoption goat domestication at Ganj Dareh was accompanied by shifts in technological organization.

This paper is organized in different sections. The first presents Ganj Dareh and the history of research at the site, including the results of prior studies. This allows us to develop working test hypotheses for what the artifactual record should indicate about human behavioral dynamics at the dawn of animal domestication. The second details the field documentation and material currently curated in the Laboratoire d'archéologie de l'Anthropocène at Université de Montréal (Canada), on which the proposed reconstruction is based. The third part of the paper presents the result of our stratigraphic reconstruction in the western part of the site, followed by a presentation of the lithic and clay object record. The paper concludes with an analysis of the shifting distribution in the density of these materials across the site's stratigraphy, a discussion of new mortuary data from the site and situates these results in the broader context of the Aceramic Neolithic in the Zagros and neighboring regions.

## Ganj Dareh: A history of research

The site was discovered and first tested in 1965 by Prof. Philip E.L. Smith [7] who would subsequently excavate about 20% of it over the course four field seasons in 1967, 1969, 1971 and

1974 [27–32]. This work revealed the presence of five distinct levels that were labeled A to E, from top to bottom, with the top of the mound having been heavily affected by erosion [25, 33]. The succession of these five levels composed a ca. 7m tall roughly circular tell that had been cut into by local villagers on its western side, and that had been partly plowed away on its northern side, giving it an asymmetrical outline. Fig 1 presents a previously unpublished profile of the succession of these five levels. Excavations proceeded following a grid of 2x2m units that were dug in semi-arbitrary horizontal increments of varying thickness. Each increment in each unit was given a unique log number, allowing its precise positioning within the mound.

The chronology of the site has always represented a thorny issue. Two charcoal samples collected in 1965 yielded age ranges of 10,400 ± 150 BP (GaK-807) and 8910 ± 170 BP (GaK-994), which were later correlated to Levels E and D, respectively [34]. This indicated that the site may have been occupied as early as the 10th Millennium BC, although all subsequent dates commissioned by Smith indicated occupations constrained to the 8th or 7th Millennium [35–37]. More recently, dates on collagen from goat remains [8] and on collagen from human remains [38] have also indicated that the GaK-807 date is aberrant, and constrain the accumulation of the site's five levels to a 200–300 year span after about 10,100 cal BP. In sum, while there has so far been no clear explanation for the presence of the aberrant date from Level E, the majority of the available evidence indicates that the site formed very rapidly in the early Holocene.

Beyond its early age, a number of rather exceptional aspects make Ganj Dareh truly unique in the panorama of Aceramic Neoltihic sites in the Zagros and of the Middle East more generally. As concern the Neolithization process itself, the site's main claim to fame is that, at ca. 10,200 cal. BP, it has yielded the earliest evidence of goat domestication in the world [8]. This is inferred from the mortality patterns of the goat assemblages from all levels at Ganj Dareh, in which sub-adult males appear to have been selectively killed off relative to females, which were killed later in life, a pattern that is consistent with that of the culling imposed on present-day goat herds raised for meat [8, 39, 40]. Given the site's location at 1400m asl in the central Zagros, it appears that the human-goat interactions that would eventually lead to the domestication of the latter were taking place at the margins of the 'Fertile Crescent' [9, 41].

While the goat remains from Ganj Dareh provide the earliest indisputable archaeological evidence of the management of goat herd by humans [8], it remains an open question as to whether or not this strategy was developed *in situ* or indeed in the central Zagros. In fact, evidence from southeastern Anatolia indicates that managed herds of goats were present in that region by 10,500 cal BP [42–45]. The fact that managed goats are rapidly found as far west as Cyprus by about 10,000 cal BP [46] and east into the central Zagros, as at Ganj Dareh where this harvesting strategy is fully in place by the same time, suggests that the origins of goat management predates the occupation of the site by at least several centuries. This, and recent paleoethnobotanical work (see below), would indicate that the Central Zagros was part of the

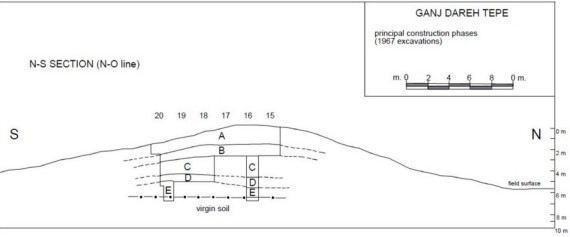

**Fig 1. Unpublished N-S profile of the excavations at Ganj Dareh in 1967.**

'round house phase' documented across the Eastern Fertile Crescent [47] that is the most probable starting point for the food production strategies that are already well-established by the time mud-brick structures were built at Ganj Dareh to enable its long-term occupation.

Goat domestication is the most solid, but not the only evidence of human food production at Ganj Dareh. The only paleobotanical analysis conducted at the site–almost 40 years ago now–was argued to show that domestic-type two-rowed hulled barley *Hordeum distichum* grew alongside its wild progenitor *Hordeum spontaneum* and that barley as a whole increases in frequency over the Ganj Dareh sequence [48]. As well, lentils argued to conform dimensionally to the wild type are documented throughout the sequence, and that no other domesticated crop was reported from the site [48]. Some years later, Charles [49] proposed that the presence of domesticated barley at Ganj Dareh was probably best explained as the result of it being grown as goat fodder, as opposed to human consumption since it conforms to the pattern of dung deposits at other contemporary sites. This is also reinforced by the fact that There is also evidence of large-scale pistachio and almond consumption at Ganj Dareh, especially in the lowest levels. Furthermore, the original paleoenvironmental reconstructions indicate that the site's original setting at the beginning of the Holocene was forest-steppe, before it was opened up by increasing fire incidence [48, 50].

However, these early data lacked the context now provided by several additional decades of additional paleobotanical research. This research, notably at sites such as Chogha Golan suggest in the Zagros and the Eastern Fertile Crescent more broadly, people were actively engaging in low-level food production involving indigenous grasses and particularly barley [51–56]. Further, the size of lentils is now known to be a lagging indicator of their domestication, casting some doubt on Van Zeist et al.'s [48] interpretation that they were only present in their undomesticated form [57, 58]. Indeed, Savard and others have demonstrated that a heavy emphasis on pulses was characteristic of human subsistence in the region at least from the Epipaleolithic on [59–61]. This indicates a very long history of people managing these plants in the region, and perhaps even that they were independently domesticated. Likewise, analyses of recently collected samples from Ganj Dareh indicated that seed generically interpreted as 'Triticoid type' by Van Zeist et al. [48] in actuality likely belong to *Heteranthelium piliferum*, which would have been part of a complex of wild grasses systematically harvested by the site's occupants [62, 63]. Thus, a radically different picture about plant domestication at Ganj Dareh and in the Eastern Fertile Crescent more generally emerges from recent investigations relative to that of the 1980s. This underscores the need to provide proper context to important, but poorly known sites excavated more than a generation ago, such as Ganj Dareh.

Another defining character of Ganj Dareh is its architecture. While Level E has been described as devoid of any permanent structures, it comprised "at least thirty round or ovoid "fire pits" or basins dug into virgin soil. . . [s]ome . . . filled with stones while others are empty. A few were used more than once and contain stratified deposits" [25]. In contrast, the overlying four levels, and in particular Level D, "contains solid architecture of various kinds" that "provide an argument that after Level E, Ganj Dareh was something more than a temporary or seasonal site" [26]. The 'firepits' of Level E are associated with conspicuous amounts of pistachio (and to a lesser amount almond) nutshells, which suggests they may have been used when these were available in the mid-summer [48]. In contrast, in the architectural levels, "the buildings are constructed of various combinations of clay, mud-brick, and wood. Very little stone was incorporated in spite of the abundance of limestone in the immediate surroundings, and there are no stone wall foundations or extensive areas of pavement or flooring. Mud plaster was lavishly used for floors and walls" [26].

The site's other artifact assemblages are also very rich. They comprise abundant chipped stone tools ranging from rough flakes and debris to fine blades and bladelets struck from

prepared cores; some of the blades also bear the edge sheen characteristic of lithics used as sickle components or to harvest plants [64–68], although these data are based on studies comprised of very small samples of lithics (1640 to 3179 pieces, out of many tens if not hundreds of thousands) and on preliminary observations published as part of field reports (e.g., [7, 25, 27, 30, 31, 33]). Ground stone tools such as mortars, querns and pestles are also present [64], as are bone tools of various sorts, including notched goat scapulae, awls, smoothers (*lissoirs*), flensers, ginners and rods [69, 70].

Another distinctive element of Ganj Dareh's material culture are its rich and differentiated clay artifacts. These include human and animal figurines [71, 72], ceramic vessels such as small vases [73], and a wide array of clay tokens, including spheres, disks and cones [74], of which only a subset of distinctive 'gashed cones' have so far been summarily described [75]. Analyses of the firing temperatures of some of the sherds from the various levels at Ganj Dareh indicate poorly-controlled baking, as reflected by the highly variable conditions resulting from baking over open fires and by occasional large-scale conflagrations, which probably destroyed the village in Level D [25, 26, 30, 33, 76].

Many human remains have also been recovered from all levels at Ganj Dareh, often as part of burials. In total, while 41 individuals were reported in preliminary excavation reports [30, 31], mostly from burials, Merrett's exhaustive analysis of the human remains from Ganj Dareh [77] identified a total of 116 distinct individuals represented by as little as single elements to nearly complete skeletons. Some of the buried individuals bear evidence of cranial deformation [78–80], while overall health conditions appear to have been rather good, with low incidence of cavities and occasional traces of porotic hyperostosis, likely caused by zoonotic brucellosis caused by sustained contact with ovicaprids [77, 81, 82]. Recent analyses of stable carbon, nitrogen and sulfur isotopes on 20 individuals indicate that the Neolithic occupants of Ganj Dareh all shared a diet largely based on $C_3$ plants, that subadults may have been weaned using supplement with distinct carbon values and that one of the older male individuals may have been a transhumant shepherd [83]. Genetic data have also been obtained from a few individuals, with Gallego-Llorente et al. [84] concluding that the Ganj Dareh population was more closely related to hunter-gatherer groups from the Caucasus than to contemporary groups from Anatolia, suggesting an independent development of agriculture in the Zagros. Drawing on a larger sample of ancient Near Eastern genomes, Lazaridis et al. [85] confirmed the genetic differences between the earliest agro-pastoral populations at the western and eastern ends of the Fertile Crescent (see also [20]).

In spite of this wealth of data, much of it remains preliminary or based on descriptions of small and/or unrepresentative samples. Further, in spite of work at the site having concluded more than 45 years ago, a number of key elements make a holistic understanding of the site difficult. This remains true in spite of recent targeted reexcavations of parts of the site, which highlight the difficulties of linking contemporary observations to those of Smith [15]. Foremost is the fact that, while it has often been stated that the site is comprised of five levels, no detailed stratigraphy or sedimentary description has ever been published. This is all the more important since tell deposits are notoriously depositionally complex and a five-level scheme is unlikely to capture the true complexity of a site as extensive as Ganj Dareh. In fact, Smith himself [26] has stated that Level D was likely composed of multiple occupations. Additionally, while it has repeatedly been argued that Level E was occupied seasonally by mobile foragers, while Levels D-A were occupied by more sedentary food producers, this is mostly based on the presence of architecture and general qualitative statements about changes in material culture.

This paper presents new data to clarify some of these questions. This new information was obtained as a result of the rediscovery of the Ganj Dareh archives housed in the Département d'anthropologie at Université de Montréal, which had been exported by Smith following his

excavations at the site in the 1960s and 1970s. Authorization to study this material was provided by the Faculty of Arts and Sciences of Université de Montréal under whose authority it is being curated at the university. Beyond lithic, clay and bone artifacts, this material also includes all of the original field documentation, including excavation forms, field notes, planimetries, stratigraphic profiles, in addition to a synthetic record of all excavation units and summaries on filing cards of each unit's artifact content. In 2017, this material was rediscovered and moved out of storage into the collections of the Laboratoire d'archéologie de l'Anthropocène directed by the lead author, who initiated an assessment of its integrity and scientific potential. Here, we present the first results of these endeavors, which were undertaken in order to test whether:

1. the primary field documentation allowed us to reconstruct the site's stratigraphic profile and whether that stratigraphy was limited to five depositional levels;

2. the expectation that the planimetry of at least some of Ganj Dareh's occupation levels could be reconstructed, as suggested by the maps presented in Smith [26];

3. the shift to a greater reliance on goat domestication over time was accompanied by significant shifts in mobility detectable in other segments of the archaeological record (e.g., lithics, ceramics);

4. decreases in mobility correlated with greater investment in technological innovation and permanent architecture;

5. more sustained and lengthier occupations were accompanied by other behaviors linked to place-making and/or ritual (e.g., burials, shrines, altar).

## The stratigraphy of Ganj Dareh

As mentioned, in recent years, it has proved challenging to integrate the data from Ganj Dareh into larger debates about the origins of animal domestication in the eastern Fertile Crescent. In large part, this was due to the absence of a published stratigraphy that would have helped, among other things, to understand site formation processes and to clarify the stratigraphic position and relation of the dozens of samples dated by radiocarbon over the years. Sketching out the stratigraphic context of the site was thus our first priority.

Prior to our reanalysis of the primary excavation documentation, the only information on the characteristics distinguishing Ganj Dareh's five levels had been presented in several preliminary reports [25, 27–33, 86] and in a detailed analysis of Layer D's architecture [26]. It is worth noting that these descriptions were fundamentally based on the kind of architecture contained in each level, rather than on the specific geoarchaeological and/or sedimentary characteristics of each stratigraphic horizon, which complicated the understanding of site formation processes at Ganj Dareh. The only geological information available refer to the fact that the sediments in the western portion of the site were 'harder' than those in the central and eastern parts of the site [30] and that, overall, the site was composed mostly of "very hard, gritty calcareous sediments [that] inhibited preservation" of some organic artifacts and ecofacts [33]. Needless to say, an important component of clay in the architectural levels is also expected.

Level A is defined by the presence of solid architecture, built with small rectangular red bricks and, less frequently, mudbricks, all of which are very altered. Plastered floors are also documented. The integrity of this thick (1.5-2m) level was compromised by leaching, freeze-thaw, water infiltration, erosion, and bioturbation in the form of root activity and animal burrows. This is in addition to recent intrusive burials, which recent dates have confirmed to be of

Islamic age [38]. In the central part of the site, this level yielded the remains of a kiln containing several layers of ashy deposits, as well as of a cist-like structure composed of edge-lain stone slabs. This is the only evidence of stone being used as a building material, as no stone wall foundations were documented. Smith [29, 33] mentions that this level likely comprised more than a single occupation as it comprised several sub-levels of architecture.

Level B is defined by the presence of architecture in the form of mudbrick- and *chineh*-walled houses composed of quadrangular rooms containing hearths and occupation debris and whose walls were covered by white plaster. Most of the rectangular rooms contained multiple occupation floors. Level B is described [30] to be a lateral continuation of Level C in some parts of the tell (especially the northwestern part).

Level C appears mostly restricted to the northwestern part of the site. It is thinner overall than Level B and is best represented by a cluster of a few buildings near the center of the tell. It has been described as a vertical continuation of underlying Level D into which some building foundations were dug. It comprises structures largely comparable to those of Level B, namely repeatedly occupied rectangular houses whose mudbrick and *chineh* (i.e., packed mud) walls were often coated in white plaster. The rooms composing these structures often contained hearts and several occupation layers.

Level D is the most substantial level at Ganj Dareh, both in terms of its thickness and horizontal extent. It is composed of buildings (occasionally two-storied) and several meters of burned rubble. Much of it is very well preserved, apparently due to the effect of an intense fire that raced through it and baked much of the originally unfired clay objects and structures it comprised (including artifacts). It is composed of a dense agglomeration of structures built with sun-dried mud bricks, large unfired plano-convex bricks and packed mud, and most surfaces where thickly coated with mud plaster layers. While stones were occasionally incorporated into walls, no stone foundations are documented. The two-storied structures comprised "a living surface supported by wooden beams overlying in some cases the small alcoves or cubicles" [26, 30], the latter of which were filled in by collapsed debris from the upper floor following the fire(s) that destroyed the village. The small exiguous spaces of the first floor may have been storage structures, as suggested by the presence of large clay jars, clay rimmed grinding stones and clay-slab separations. In addition to these, a number of kilns were also identified in Level D, as was a possible 'ritual niche' comprising two stacked ovicaprid skulls in the side of one cubicle located almost completely at the center of the tell.

As concerns Level E, while considering it a single level, Smith repeatedly distinguished between its 'upper' and 'basal' components. The 'upper' part has been described as a ca. 0.5-1m-thick accumulation of dark soil composed of burned earth, ash and small stone fragments, and rich in burned bone and charcoal. This part of Level E was often cut into by wall foundations in Level D. The 'bottom' part of Level E is characterized by numerous circular and ovoid pits dug directly into the 'virgin soil' on top of which the site was formed. These depressions which were 0.8–1.7m across and up to 0.5m deep generally contained limestone cobbles and chunks and were sometimes completely filled by them. Because they were also comprised of ash, charcoal and traces of burning, Smith labelled these features 'firepits.' He exposed about 30 firepits during his excavations, extrapolating that they must have numbered in the hundreds over the entire surface of the site.

Finally, while Smith never described the 'virgin soil' which composes the valley floor over which the site accumulated, he did mention the presence, several meters below it, of a gravel level containing some stone tools and an abundance of (natural?) chert nodules/fragments, which were identified in a single test pit. He interpreted these deposits as either colluvial in nature or "gravels from an Upper Pleistocene stream terrace formerly present in the valley" [86].

We were able to reconstruct Ganj Dareh's overall stratigraphic sequence by digitizing the master list of excavation units from Smith's 1967–1974 excavations and imported the resulting data into a GIS. This allowed three things. First, it permitted the first detailed reconstruction of the site's excavated area in plan view (Fig 2). This plan provides the first formal description of the excavated area at Ganj Dareh beyond an unpublished site map included in the PhD thesis of D. Merrett. This plan view also clearly shows the 'central area' which has been the focus of most of Smith's excavation, along with the precise position of the 'west' and 'east' trenches excavated in 1971 to get a sense of the extent of Level D and the stratigraphy away from the central area [30, 32, 33]. Further, we were able to confirm that the site grid was composed of 2x2m square planimetric units organized according to a traditional alpha-numeric system.

Second, it allowed us to reconstruct a dynamic 3D model of the excavated area of the tell (Fig 3 and S1 Video). This model was created by georeferencing every one of the 1602 excavation units catalogued by Smith between 1967 and 1974 and attributing them to the level they were assigned to. Beyond creating an appealing visual to present an overall view of the excavation area, this model permits a detailed reconstruction of the extent over which different levels were explored. It also provides a tridimensional framework into which individual features (e.g., burials, hearths, dating samples) can be inserted, in order to capture the stratigraphic and planimetric relationship between these features and the phase of occupation they belonged to. In the future, this will offer the possibility of precisely assessing the stratigraphic position of, say, dated samples which so far have largely been assumed to be from the same context by

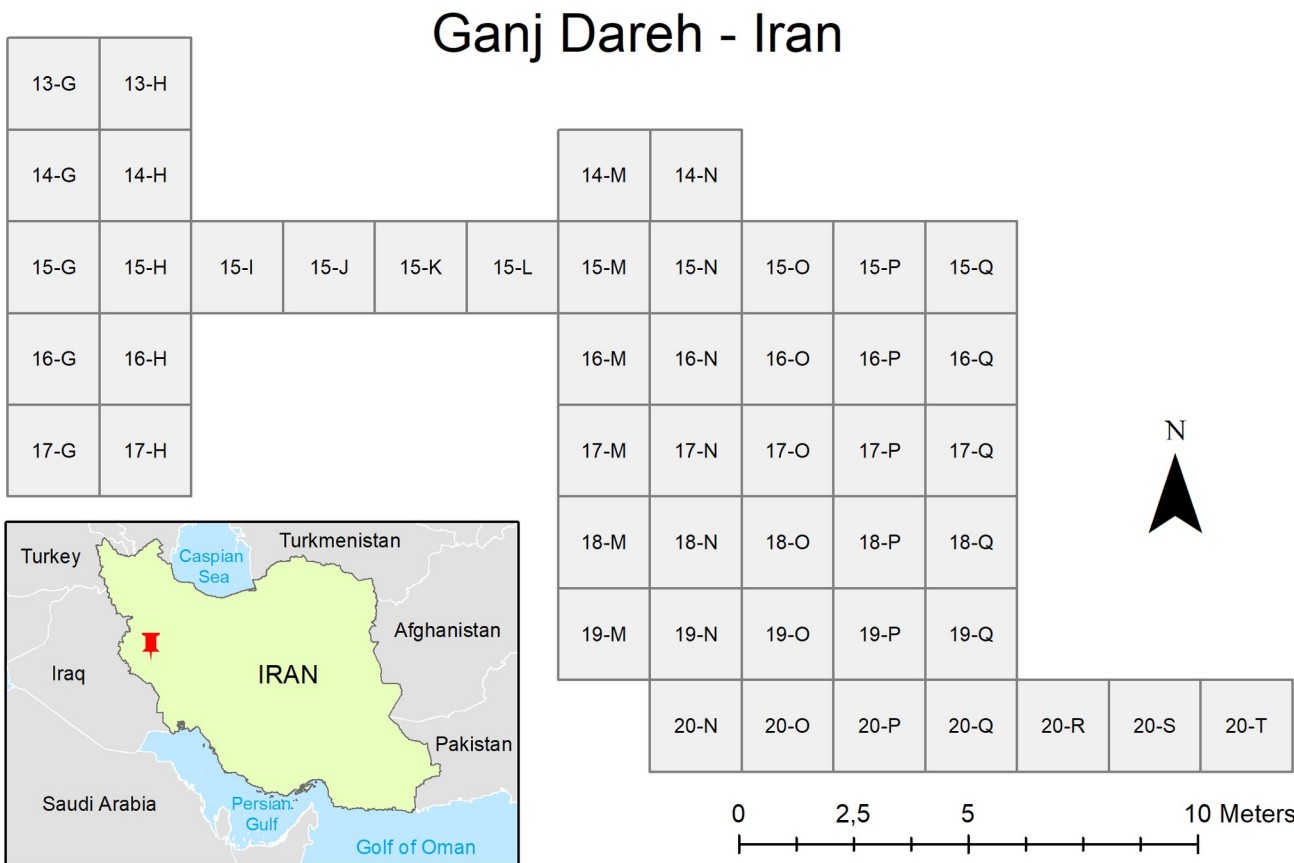

**Fig 2. Geographical position and plan view of the excavation area at Ganj Dareh.**

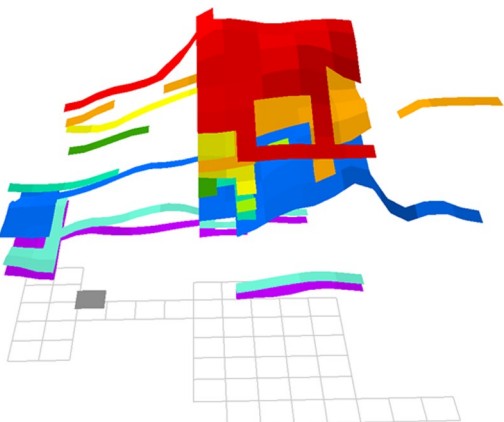

**Fig 3. Tridimensional GIS model of the distribution of Smith's levels at Ganj Dareh.** The supplementary figure includes an animated version of this model.

cross-referencing depth provenience and archaeological level although it is clear from Fig 3 that in itself, depth or level alone can be deceptive reference variables. Precisely positioning dated samples will in the future permit more complex analyses such as Bayesian modeling which may help constrain the actual occupation span of Ganj Dareh's various levels.

The GIS model of the site also allows the creation of elevation maps for each of the main occupation levels, which permits a better understanding of the site's shifting occupation dynamics over time (Fig 4). Looking at Level E (Fig 4E), we can see that this basal level was mostly exposed in the northern part of the excavation area and that it was relatively flat. In contrast, Level D (Fig 4D) was conspicuous over the entire excavated area, with particularly intense occupation towards the center of the mound; it was also noticeably thicker, with a pronounced slope from the center of the tell to its western edge. Level D thus appears to have been the first human occupation to fundamentally condition subsequent occupations that would build next to it before building above it. In fact, Level C (Fig 4C) demonstrates this trend, being restricted to the western portion of the site, which shows that after Level D, continued human settlement was constrained by the remnants of prior occupations, which is particularly noteworthy considering the original observations that some of the buildings in Level D were multistoried [26, 31, 33]. It thus appears that the occupants of Level C 'leaned' their new buildings (see below) on the burnt and abandoned structures of Level D which were thus likely incorporated and negotiated in the daily life of the site's Aceramic Neolithic inhabitants, prompting a new kind of relationship to their perceptible world and their past beyond that caused simply by a shift to sedentism (cf. [87–89]). Subsequent to this, the topographical 'leveling' effect of Level C appears to have allowed the occupation to shift back towards the center of the mound in Level B (Fig 4B) before shifting further to the south and west of the central excavated area in Level A (Fig 4A).

These data thus establish the presence of diachronic shifts in the site's occupied area over time and serve to demonstrate Smith's impression that Level C was restricted to the western part of the mound. During their targeted reexcavations in 2017 and 2018, Darabi et al. [15] also observed that the occupation of the site shifted west from the center of the tell to its periphery, showing a nice convergence between Smith's original's interpretation, the field documentation marshalled in this study, and more recent assays at the site.

Third, and most importantly in the context of the present paper, these data also allowed us to target an area to focus on for a first attempt at reconstructing the detail of the site's

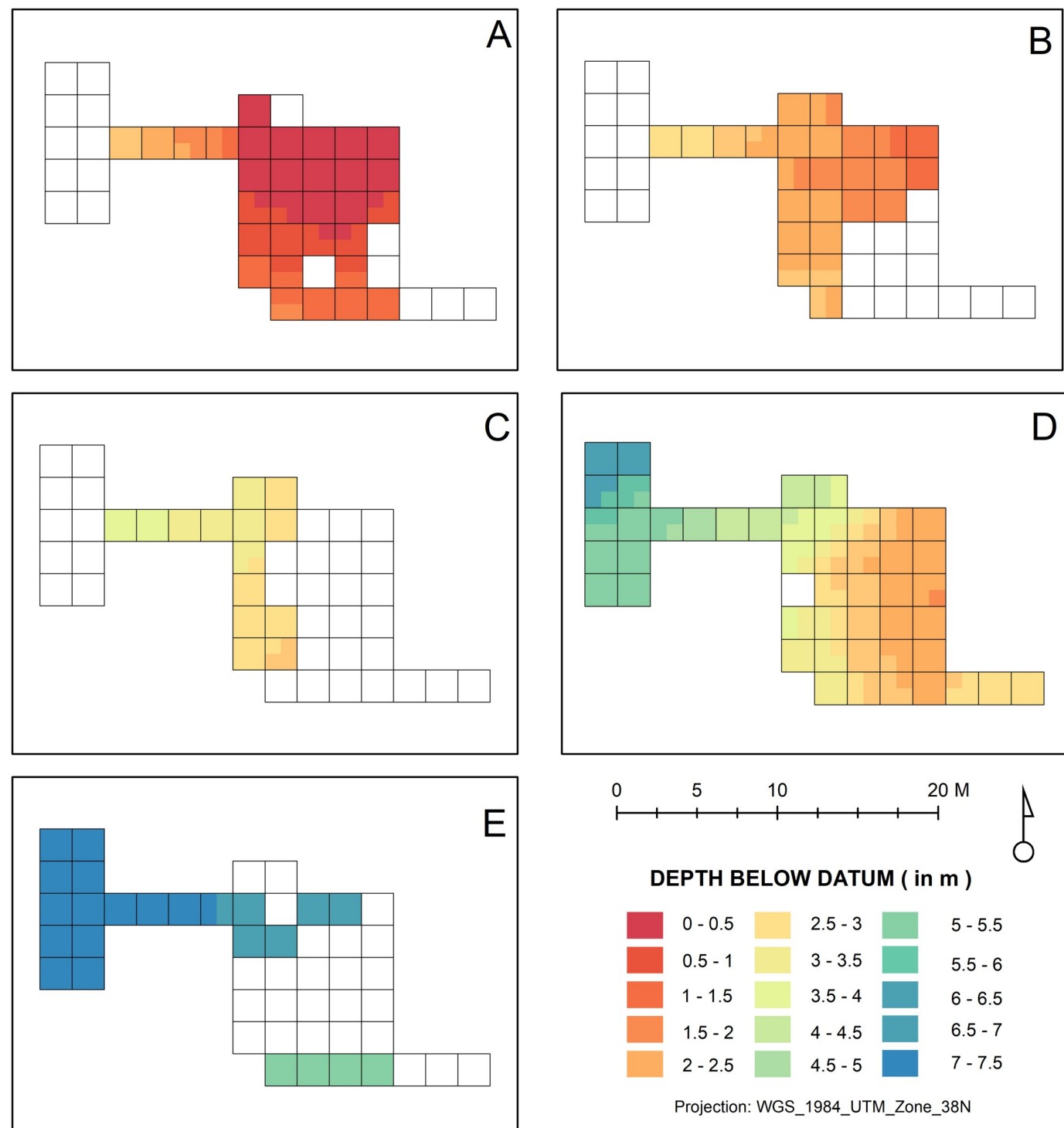

**Fig 4.** Horizontal extent of Smith's levels A, B, C, D, and E. Elevation gradient is from warmer (higher elevation) to colder colors (lower elevations), in all cases.

stratigraphy, including all of the variability reported within Smith's broad levels. This was particularly important, since the complexity of the site's stratigraphy (and its impact on prior efforts at compiling an overall stratigraphy) has been highlighted on several occasions by

Smith himself [33]. Likewise, as detailed above, Smith underscored that some of Ganj Dareh's five levels could be further subdivided, though no publication acting on this was ever produced [33]. Finally, the presence of test pits outside the main excavation area mentioned evidenced in Merrett [77] and described in preliminary field reports [30, 32, 33] was a further selection factor. For instance, it had been reported that "[i]nside the mound another test pit carried to several metres below virgin soil revealed a thick gravel concentration containing many chert nodules and fragments and a few worked pieces of vaguely Middle Palaeolithic type" [32]. This raised a further element that should be included in a synthetic stratigraphy of the site.

Thus, to control for horizontal variation in Ganj Dareh's stratigraphic sequence, it was judged preferable at this stage of research to focus on one of the two trenches excavated in 1971 (i.e., the 'east' and 'west' trenches to the southeast and the northwest of the main area, respectively). A first review of the field documentation also further revealed that one of the excavation units of the west trench was where a sondage in the "Palaeolithic" deposits was made. This led us to select the 'west trench' as the area on which to base a first effort to reconstruct the site's stratigraphy. Comprising planimetric units 15-I, 15-J, 15-K and 15-L, the west trench is also 8m in length, as opposed to the east trench's 6m length, making it the longest of the two trenches, allowing us to better assess the extent some of the site's lateral stratigraphic variation. Finally, it had been reported (and documented in our GIS model) that Level C was best documented in the western part of the site, meaning that focusing on the west trench would yield the best insight into that layer which follows the most intense use of the site as a village [30, 32]. The GIS model drawn from the log of all excavation units further (Fig 3) revealed that all five of Smith's main layers were documented in the 'west trench', confirming its appropriateness for a first elaboration of the site's complete stratigraphic sequence.

## A revised stratigraphy for Ganj Dareh's 'West Trench'

Based on a review of the field documentation contained in the Ganj Dareh archives, we were able to subdivide the stratigraphy into fourteen distinct units, which were labeled and distinguished as follows (Fig 5). Our nomenclature incorporates reference to Smith's original levels, with finer subdivisions indicated by numeric designations.

### Level A-01 (equivalent to Smith levels A and /AB)

Mixed layer of inverted stratigraphy formed by human and natural action. Level A-01 is composed of sandy clay with a loose compaction and a light greyish-brown hue. Mixed sandy clay patches with reddish and yellowish hues occurred in planimetric units 15-I and 15-J. Occasional fragments of charcoal were visible throughout the deposit, while occasional loose mudbricks and large sub-angular stones were visible in planimetric units 15-K and 15-L. A hearth cuts into A-01 in the northeast corner of planimetric unit 15-I. This feature represents the latest phase of human activity in the study area.

### Level B-01 (equivalent to Smith level B)

Sandy clay deposit of hard compaction with a greyish-brown hue. Occasional loose mudbricks and charcoal fragments were visible throughout the deposit. Level B-01 was cut by a hearth in planimetric units 15-K and 15-L.

### Level B-02 (equivalent to Smith level B/C)

Clay deposit of firm compaction with a brownish hue. Occasional loose mudbricks were visible throughout the deposit. Since B-02 overlies a series of structural features that cut into C-01

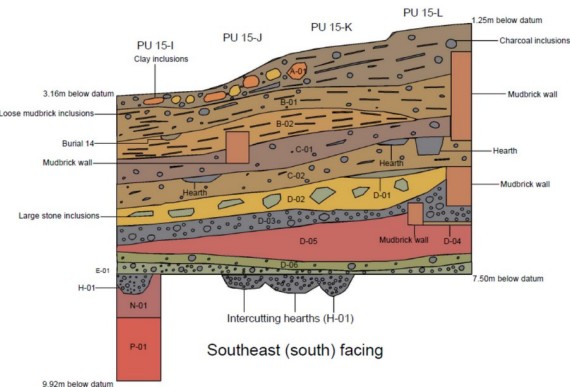

**Fig 5. Stratigraphic profile of the West Trench, Ganj Dareh.**

below, this deposit may have been formed by demolition or structural collapse. B-02 was cut by a child burial in planimetric unit 15-I. Artefacts associated with the burial included a single flake core and animal bones. The child burial was orientated south to north. The child was estimated to have died at approximately 4 years of age.

### Level C-01 (equivalent to Smith level C)

Clay deposit of firm compaction with a brownish hue and occasional charcoal inclusions. Level C-01 appears to represent a major construction phase, since it was cut by five walls across all planimetric units and a multiple burial containing the remains of three individuals in planimetric unit 15-I. One of these mudbrick walls shares a stratigraphic relationship with the multiple burial. Considering this relationship, this wall was initially interpreted as a sarcophagus in the original site archive. However, closer inspection suggests that the burial cuts into the floor surface of a subterranean building. The multiple burial was orientated from northeast to southwest.

### Level C-02 (equivalent to Smith level C/D)

Firmly compacted layer with a brownish hue and occasional fragments of charcoal. Level C-02 was cut by three hearths spanning across all four planimetric units.

### Level D-01 (equivalent to Smith level D)

Highly compacted layer with a greyish-brown hue containing significant quantities of lime. This layer only occurred in planimetric units 15-K and 15-L and appears to have accumulated against three mudbrick walls, which cut into Level D-02 below. This layer was interpreted as a plaster floor surface in the primary documentation, which is probably accurate given the presence of lime in the deposit.

### Level D-02 (equivalent to Smith level D)

Loosely compacted layer with a yellowish-red hue and frequent large stone inclusions. Level D-02 appears to represent a major construction phase, since it was cut by three mudbrick walls.

### Level D-03 (equivalent to Smith level D)

Loosely compacted layer with a blackish-grey hue and an abundance of charcoal. It is likely that Level D-03 demarcates the well documented burning event associated with level D. Level D-03 was cut by a hearth and a refuse pit in planimetric units 15-K and 15-L.

### Level D-04 (equivalent to Smith level D)

Highly compacted layer with a greyish brown-hue. This layer was no more than c. 0.05 m in depth and was only discernible in planimetric unit 15-L. Level D-04 was situated within the confines of a mudbrick wall. Level D-04 may therefore have been either a floor surface or a layer of trample, formed during the construction the mudbrick wall against which it accumulated. Given that Level D-04 was cut by a burial to the north and east of planimetric unit 15-L in planimetric units 14-M and 15-M respectively, it is likely that Level D-04 was a floor surface.

### Level D-05 (equivalent to Smith level D)

Highly compacted layer with a reddish-brown hue. This layer was cut by a mudbrick wall. Level D-05 is cut by the earliest construction phase visible in the study area.

### Level D-06 (equivalent to Smith level D)

Loosely compacted layer with a greenish hue. Fragments of charcoal and burnt stone were visible throughout the deposit.

### Level E-01 (equivalent to the top part of Smith level E)

Loosely compacted layer with a greyish hue. Frequent charcoal fragments and occasional large burnt stones were present throughout the deposit. Level E-01 accumulated above the virgin (natural) soil horizon that was cut by several ovoid 'firepits' (hearths).

### Level H-01 (equivalent to the 'firepits' at the base of Smith level E)

Several large ovoid hearths cutting into a natural deposit. The fill of these hearths had a greyish hue with a high charcoal content.

### Level N-01 (equivalent to Smith's 'virgin soil')

Highly compacted natural soil with a reddish-brown hue. This horizon was cut by several large ovoid hearths. Some of these hearths appear to have been lined with stones, which may have been placed there to enable their makers to ignite fires before they were extinguished by wind.

### Level P-01 (no equivalent in Smith's stratigraphy)

Colluvial deposit of hard compaction with a brownish-red hue. Several apparently (Middle?) Paleolithic artefacts were found in Level P-01. This deposit was excavated in a 1 x 1 m sondage (test pit) in the NW corner of planimetric unit 15-I to the limit of excavation at -9.92 m.

### Planimetry and horizontal organization

The level of detail in the 1967–74 field documentation allowed us to reconstruct the planimetry of all the levels identified in our renewed stratigraphy. Some of these were limited to mapping out the distribution of hearth features (e.g., Levels A-01, B-01 and C-02) or of individual burials (e.g., Level B-02). Others were much more informative and yield unprecedented resolution on shifts in the spatial organization of human activities at Ganj Dareh over the course of the site's occupation. While we return to the burials recovered in Levels B-02 and C-01 in a separate section below, this section describes the most complex levels we were able to identify.

**Level C-01 (Fig 6).** This is the most complex architecture-bearing level identified in the West Trench; not coincidentally, this is one of the thicker units documented in this part of

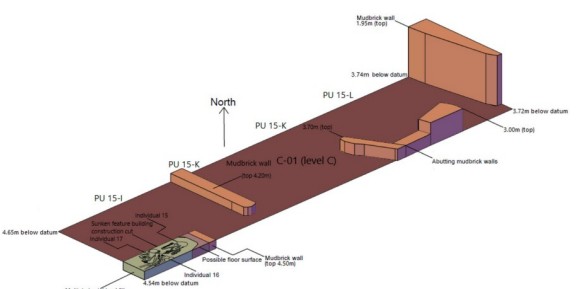

**Fig 6. Planimetry of level C-01.**

Ganj Dareh. In it, a number of architectural structures have been identified, most notably a mudbrick wall over 1.70m in height at the eastern edge of the sampled area, that was likely part of a multistory building, which cuts into underlying Level C-02 (Fig 5). Other structures include partial mud-plaster floor surfaces and free-standing as well as abutting ensembles of mudbrick walls. One concentration of mudbrick walls and floor surfaces at the southwestern corner of the sampled area comprises a triple burial that appears to have been deposited in a compact subfloor deposit and to have reused wall foundations as part of its enclosure (see 'Burials' section below). The presence of two-storied buildings with stout subfloors alveolar enclosure in this part of the site is reminiscent of what has been described as the architectural model of Level D by Smith [26]. It also confirms that the most substantial occupation of the site following the fire that ravaged Level D shifted westward (Fig 4), as well as the pronounced east-to-west slope of the deposits on the western side of the tell.

**Level D-02 (Fig 7).**  The mudbrick architecture comprised in this level includes the base of a series of abutting wall that outline roughly rectangular ca 1x1.5m spaces whose size and morphology recalls those of the buried storage spaces documented in the central part of Ganj Dareh during its most intensive phase of occupation [26]. This indicates that the structures documented and mapped in the central excavation area for Level D extended only slightly to the west of it. This interpretation is bolstered by the floorplan from level D-04 (Fig 8) that also includes mudbrick architecture and a compact floor surface but only in planimetric unit 15-L, immediately to the west of the central excavation area. The only other structures documented for 'Level D' in the West Trench are found in Level D-03 (Fig 9): they include a 1m-wide hearth filled with subangular stones and, immediately to the south of it, a ca. 35cm-deep pit capped by a series of stone slabs.

Perhaps the most striking planimetry we were able to reconstruct is that for Level H-01, which corresponds to Smith's so-called 'firepit' level (Fig 10). Some prior publications have shown oblique views of some of the firepits, but this is the first plan-view *in extensio* illustration of this unique level. In contrast to architectural levels D-01 to D-06, the firepits are

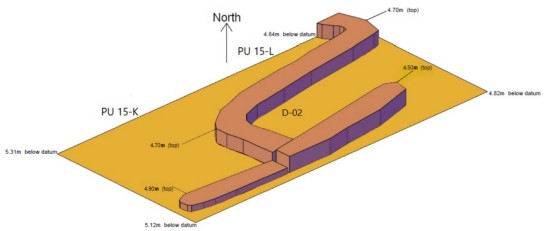

**Fig 7. Planimetry of level D-02.**

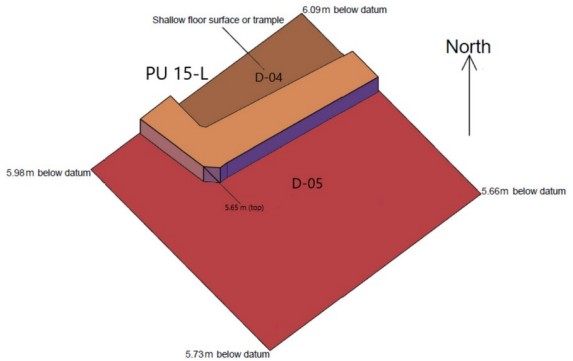

**Fig 8. Planimetry of levels D-04 and D-05.**

conspicuous in the deposits west of the central excavation area and the 12m² area of Level H-01 exposed in the West Trench comprises parts of at least ten 'firepits.' These are shallow round-to-oval depressions that range from 17cm to 43cm in depth (most being around 30cm deep) and that were dug directly into the 'virgin soil' found below the tell (i.e., Level N-01). Most of these pits included large stone or cobble clusters at their bottom, in addition to being filled with ashy sediments that were rich in charcoal and which Van Zeist et al. [48] indicate comprised a large amount of pistachio and almond carbonized shell fragments. We were also able to map out the extent of the "arc of small stones slabs placed on edge" bordering one of these pits in planimetric unit 15-I, which had been mentioned in preliminary site reports [30]. This is the only structure other than the pits themselves that is documented for Level H-01 that extends almost completely along the eastern margin of the largest pit in planimetric unit 15-I.

Additionally, the field documentation allowed us to determine that one of the firepits in planimetric unit 15-J was subsequently cut by a second pit, suggesting that Level H-01 may include several phases of occupation. This could help explain why some pits are filled with burnt rocks and cobbles while others are almost devoid of them; it may be that cobbles from earlier pits were reused to line the bottom of later ones. While this remains to be confirmed, it could provide a first element to identify the loci of the very earliest occupations of Ganj Dareh. A last observation about Level H-01 concerns the dimensions of the pits. In the area under consideration here, none are larger than 1.7m in diameter and most are considerably smaller, about 1m in diameter. This, combined with their contents, strongly suggest that the 'firepits' were in fact combustion structures rather than the bases of sunken structures, such as semi-subterranean houses dwellings.

## Ganj Dareh artefact assemblages: Lithics

To date, no comprehensive analysis of the lithic assemblages from Ganj Dareh has been conducted. Techno-typological analyses of small samples of stone tools stored at the National

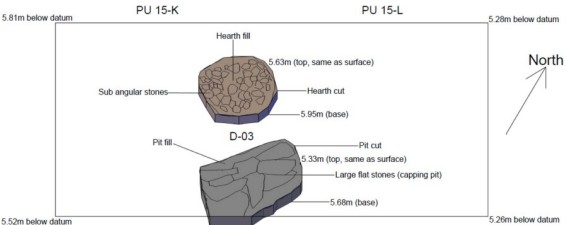

**Fig 9. Planimetry of level D-03.**

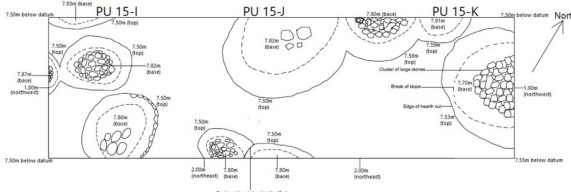

**Fig 10. Planimetry of level H-01.**

Museum in Tehran were recently conducted [66–68]), concluding that Ganj Dareh's lithics show Early M'lefaatian affinities, in particular to the Kermanshah group of that industry [90, 91]). This diagnosis appears to be borne out by the samples recovered in recent fieldwork [15]. In this, the lithic assemblage from Ganj Dareh appears comparable to assemblages from contemporary sites like East Chia Sabz [13, 92] and Asiab [16]. However, Nishiaki [66] highlights that the Ganj Dareh assemblage is also unique in comprising sickle blades and backed points, and wells as in showing distinctive bladelet production strategies. It bears emphasizing that none of these analyses have indicated the presence of inter-level lithic differences between the five levels at Ganj Dareh and have considered the Tehran sample as representative of lithic production for the site as a whole.

Our reanalysis of the Ganj Dareh collections and archives at Université de Montréal allow us to shed important new light on some of this internal variability. Indeed, the Ganj Dareh archival material also includes records of the artifactual content of each excavation unit. These records (index cards) were digitized and their information was extracted and incorporated into a centralized database that allowed us to link every excavation unit to its artifactual content and thus to the Ganj Dareh level they belonged to. In the 'West Trench' the focus of the present study, by using the same excavation unit numbers as those used to reconstruct the fourteen levels of our stratigraphy, we were able to compile baseline lithic information for each level. To our knowledge, beyond an MSc thesis that reported a classification structure to analyze the lithic industry at Ganj Dareh [65], this is the first time that a comprehensive quantitative assessment of the site's lithic assemblages has been conducted.

Our data indicate that the lithic assemblage from the West Trench is represented by a total of 8980 pieces, including 116 cores and 830 retouched pieces (Table 1). Assuming for the sake of argument that this area is representative of the rest of the site, we can extrapolate that the total lithic collection from Ganj Dareh amounted to ca. 108,191 pieces. We stress these numbers, and the fact that the sample described in Table 1 comes from an area representing less than 10% of the total site area, to provide some context to three prior lithic analyses of Ganj Dareh published between 1975 and 2016. Pullar [64] analyzed a sample of 1640 pieces (comprising 527 tools and cores and 1013 waste flakes) from levels A to D collected from a single but unspecified 2x2m planimetric unit. Thomalsky [67, 68] analyzed a sample of lithic implements stored at the National Museum in Tehran but did not specify the number of pieces analyzed. However, this is a sample size was presumably similar to that analyzed by Nishiaki [66], which comprised 3179 pieces, including 2464 pieces of debitage and 714 tools and cores, with only 1939 of the total coming from stratified contexts. Contrasted to our data from the West Trench, this implies that prior analyses of stratified lithic samples were based only on 1.64% to 1.93% of the total assemblage, underscoring the critical need for detailed holistic analyses of the site's lithic assemblages to accurately characterize its chipped stone technology. Such efforts are currently underway under the supervision of the lead author.

**Table 1. Lithic counts for each of the levels identified in the West Trench (Fig 5).** Level N-01 is the natural 'virgin soil' devoid of traces of human activity and is thus excluded.

| Stratigraphic horizon | Number of retouched lithics | Number of Cores | Total number of lithics | Percentage retouch | Lithic volumetric density (pcs/m$^3$) |
|---|---|---|---|---|---|
| A-01 | 86 | 14 | 1316 | 6.53 | 80.64 |
| B-01 | 69 | 13 | 1211 | 5.70 | 102.28 |
| B-02 | 31 | 12 | 688 | 4.51 | 119.44 |
| C-01 | 55 | 8 | 817 | 6.73 | 88.04 |
| C-02 | 98 | 23 | 1311 | 7.48 | 130.06 |
| D-01 | 19 | 2 | 166 | 11.45 | 171.13 |
| D-02 | 172 | 12 | 1192 | 14.43 | 158.51 |
| D-03 | 76 | 14 | 767 | 9.91 | 106.53 |
| D-04 | 98 | 4 | 728 | 13.46 | 53.53 |
| D-05 | 0 | 0 | 0 | 0 | 0 |
| D-06 | 14 | 5 | 160 | 8.75 | 52.63 |
| E-01 | 86 | 9 | 551 | 15.61 | 137.75 |
| H-01 | 18 | 0 | 51 | 35.29 | 5.77 |
| P-01 | 8 | 0 | 22 | 36.36 | 15.49 |

The size of the sample described in Table 1 and the fact that it is distributed across multiple distinct levels also permits an independent evaluation of the idea that the lithic assemblages from different parts of the sequence are homogeneous in terms of their technological organization. What is immediately apparent is that artifact densities appear to vary widely across the 14 stratigraphic levels in our sequence. Likewise, there appears to be a notable degree of variation in the frequency of retouched pieces. In other studies, these two variables (i.e., lithic volumetric density and retouch frequency) have been used jointly to highlight different facets of technological organization using what has been called a 'whole assemblage behavioral index' and can be used to track change in lithic management systems and land-use strategies [93–100]. Importantly, these studies establish that this method is no more influenced by sampling bias than other methods of lithic analysis, including typo-technology. While the method has mostly been used in Paleolithic complexes, it has also shown to be useful in analyzing Epipaleolithic, Mesolithic and early Neolithic assemblages [99, 101, 102]. Given this and the assumption that Ganj Dareh shows a shift from a forager adaptation in units E-01 and H-01 (corresponding to Smith's level E) to a more sedentary one in the overlying levels, we judged that this method would be a good way of testing whether this is reflected by another dimension of the archaeological record, besides the presence/absence of architecture.

From the lithic database, we thus extracted two basic values: the total number of lithics and the total number of retouched tools per stratigraphic horizon. From this we were able to calculate the frequency of retouched pieces, taken as a gross indicator of lithic curation for individual assemblages. In turn, the stratigraphic database allowed us to calculate the volume of sediment excavated for each stratigraphic unit. Dividing the total number of lithics by the volume of each stratigraphic unit allowed us to derive the second key variable for the WABI, namely, lithic density per cubic meter of excavated sediment. As detailed in other studies, we expect retouch frequency and lithic density to be inversely related, with assemblages characterized by higher retouch frequencies and lower densities being indicative of a more 'residentially,' organized, more mobile lifeway, and conversely, assemblages characterized by lower retouch frequencies and higher densities being indicative of more logistical, less mobile strategies. Based on these expectations and prior claims about the nature of human occupation in Level E, we expected the assemblages from stratigraphic units H-01 and E-01 to show more

curated assemblages diagnostic of greater residential mobility and the assemblages associated with mudbrick architecture to show a more expedient technological organization diagnostic of a more settled lifeway.

The lithic data from all assemblages conform strongly to the expect negative relationship between retouch frequency and lithic density (r = -.728, p = .00473, N = 14), but only partly bear out the stratigraphic expectations just outlined (Fig 11). While Level H-01 displays the expected pattern and clusters at the curated/'high mobility' end of the spectrum along with the 'Paleolithic' assemblage from Level P-01, all the other assemblages, including that from Level E-01, cluster towards the expedient/'low mobility' end. While this is expected from the levels with architecture, which are argued to represent sedentary occupations of Ganj Dareh, this is completely unexpected for Level E-01, which on the basis of previous interpretations was expected to reflect a more mobile forager adaptation. These data therefore reinforce the validity of the finer-grained stratigraphic subdivisions proposed above and the existence of a real distinction in the lithic technological organization and land-use strategies of Levels H-01 and E-01. To the extent that these patterns of technological organization correspond to some form of techno-typological classification, these results further suggest that more detailed formal analyses of the lithic material from the base of the Ganj Dareh sequence (i.e., Levels P-01, but especially Level H-01) could reveal some important techno-typological differences that would break with the overall M'lefaatian character of Ganj Dareh's lithic technology. It is conceivable that the lithic assemblages from Levels P-01 and H-01 could represent earlier Epipaleolithic occupations of the site that have so far been unrecognized, although this remains to be tested by future detailed techno-typological analyses. The presence of Epipaleolithic deposits at the base of Ganj Dareh would, however, align itself well with the emerging pattern of hunter-gatherers occupying key sites in the region during the climatic amelioration of the early Holocene to take advantage of increasingly abundant resources prior to the development of food production [24]. Although this picture is complicated by recent data showing a perduration of human occupations at Palegawra

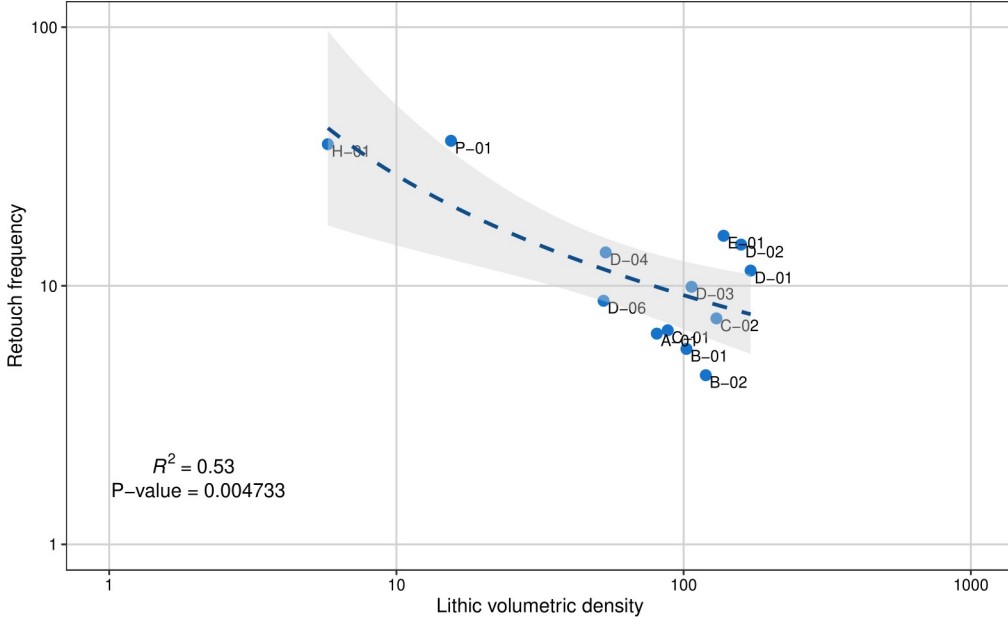

**Fig 11. WABI graph for the various levels at Ganj Dareh.**

Cave during the terminal Pleistocene [23], it would agree well with the pattern of occupation documented at nearby sites including Abdul Hosein, Asiab, East Chia Sabz, Chogha Golan and Sheikh-e-Abad [11, 16, 19, 51, 52, 92, 103].

To close the discussion on the site's lithic assemblages, our preliminary observations support prior observations made about the raw materials used to manufacture the Ganj Dareh lithic assemblages. The exploited raw materials indicate the dominance of fine- and coarse-grained radiolarian cherts (especially reddish and grey/tan varieties) similar to those available from outcrops in the region surrounding the site, suggesting local provisioning [64, 66, 104, 105]. As already noted by Smith [31], obsidian is not documented in any level at Ganj Dareh.

## Clay objects

The stratigraphic distribution of portable clay objects (i.e., figurines and tokens) is also informative. Some of the diagnostic features of the figurines have been described by Eygun [72], while some of the token forms, which include discs, spheres and cones of various sorts are mentioned in Broman-Morales and Smith [75]. By cross-referencing Smith's master list of clay objects with the west trench stratigraphy, it was possible to get a general sense of the distribution of different kinds of clay objects across vertical and horizontal space (Table 2). The first thing that can be seen is that Levels P-01 and H-01 stand out by being devoid of clay objects. In Layer E-01, clay objects were represented by two zoomorphic figurines, three geometric tokens and an undiagnostic clay object. If we exclude the assemblage from disturbed Level A-01, this is the third densest assemblage of clay objects for the entire site, indicating that clay technology appeared rather suddenly at Ganj Dareh. This makes the contrast to the clay-less underlying levels all the more dramatic. From Level E-01 up, the volumetric density of lithic and clay objects roughly covaries, likely speaking to the overall density of occupation at the site ($r^2 = 0.3393$, p = 0.02883). It is also worth highlighting that geometric tokens form the majority of all recovered clay objects in the west trench (22 or 47.5%), followed by zoomorphic figurines (18 or 29.5%). Only a single anthropomorphic figurine was recovered in the West Trench, in layer C-02.

**Table 2. Counts of clay objects for each level identified in the West Trench (Fig 5).** Level N-01 is the natural 'virgin soil' devoid of traces of human activity and is thus excluded.

| Stratigraphic horizon | Number of zoomorphic figurines | Number of anthropomorphic figurines | Number of geomorphic figurines | Number of non-geomorphic figurines | Total number of figurines |
|---|---|---|---|---|---|
| A-01 | 2 | 0 | 13 | 7 | 22 |
| B-01 | 1 | 0 | 0 | 1 | 2 |
| B-02 | 0 | 0 | 1 | 1 | 2 |
| C-01 | 1 | 1 | 3 | 1 | 6 |
| C-02 | 1 | 1 | 5 | 1 | 8 |
| D-01 | 0 | 0 | 0 | 1 | 1 |
| D-02 | 1 | 0 | 2 | 0 | 3 |
| D-03 | 7 | 0 | 0 | 2 | 9 |
| D-04 | 0 | 0 | 0 | 0 | 0 |
| D-05 | 2 | 0 | 1 | 0 | 3 |
| D-06 | 1 | 0 | 0 | 0 | 1 |
| E-01 | 2 | 0 | 3 | 0 | 5 |
| H-01 | 0 | 0 | 0 | 0 | 0 |
| P-01 | 0 | 0 | 0 | 0 | 0 |

## Chronology

Defining an absolute chronology for Ganj Dareh has proved a vexing issue since it was first discovered, due to what appear to be stratigraphic inversions, incoherent age ranges, unclear sample selection and position and occasional large discrepancies in reported ages. The various dates obtained prior to 2000 suggested an occupation span of up to 2300 radiocarbon years, although there was also no clear relationship between sample depth and age [34–37]. Recently, a battery of AMS radiocarbon dates on goat and human remains have apparently largely resolved the issue [8, 38]. These AMS dates indicate that the ca. 100-200-year overlap in age ranges across all five levels likely represents the total span of occupation at Ganj Dareh, although there are suggestions that this may be as long as 600 years [15]. Recent dates on three mouse mandibles (two from Level D, one from Level B) give the same general picture [106]. These shorter chronologies, however, have required that a discrepant date of 10,400 +/- 150 BP (GaK-807), incidentally one of the first dated samples collected in the 1965 sondage, be either dismissed as 'aberrant' or explained away as potentially dating "basal sediments underlying the site, thus contributing to the appearance of an occupational hiatus" [38].

The cross-referencing of sample provenience with the stratigraphy presented in this paper helps resolve some of these issues. First, 7/36 (or 19.4%) published dated samples from Ganj Dareh come from the 'west trench,' which as mentioned above, only accounts for about 8.3% of the total excavated area at the site. The west trench is thus overrepresented in terms of dated samples, which further bolsters the decision to choose it as the focus of a first reconstruction of Ganj Dareh's stratigraphy. The subset of dates from this context also allows us to position them stratigraphically as well as horizontally (Fig 12). Sadly, this does not resolve the apparent contradictions among the dates from different levels recently summarized by Meiklejohn et al. [38]. Indeed, the four samples recovered from Level H-01 are all much more recent than the ones recovered from Levels E-01, D-02 and C-01. This is likely because three of them (SI-923, SI-924 and SI-925) were charcoal samples dated at the Smithsonian shortly after their recovery in 1971 [36] due to issues with bulk sample collection and pretreatment. The fourth sample (Ox-2102) was a charred barley grain recovered from flotation sample 136 [48] and that was subsequently dated by AMS [37]. However, given Ganj Dareh apparently complex depositional history, the documented presence of modern and prehistoric animal burrows and the small size of the dated barley grain, it is conceivable that this grain was introduced into Level H-01 from overlying levels through post-depositional processes.

Calibrating the age of the samples from the west trench reinforces these and Meiklejohn et al.'s [38] observations (Fig 13) as this clearly shows that the non-collagen samples from this

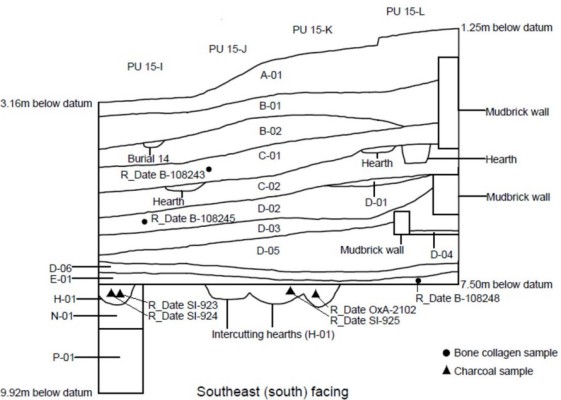

**Fig 12. Positioned dating samples located in the West Trench.**

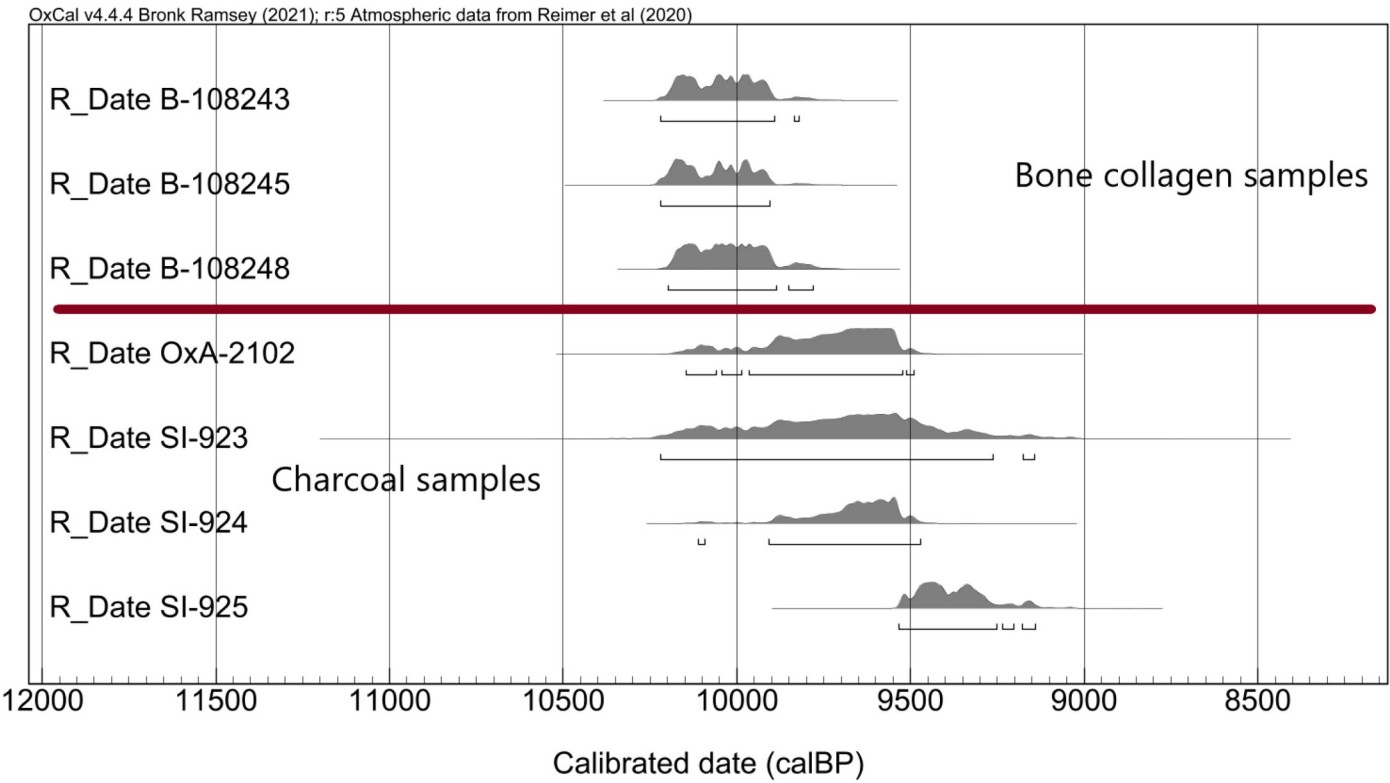

**Fig 13. Calibrated radiocarbon ages of the positioned samples in the West Trench.**

area systematically underestimate the age ranges of collagen samples by as much as 500 years. As such, it is probably safer to disregard them in analyses of the site's chronology [8, 38], meaning that we can assume to only have reliable dates for levels E-01, D-02 and C-01. Doing so confirms that bone collagen dates do not distinguish the age of the level within the range of error, reinforcing that the 11 units comprised between Levels E-01 to A-01 likely accumulated over no more than about 200 years.

This critical review of the evidence has one more important implication: Level H-01, which corresponds to the level of firepits excavated into the virgin soil, remains undated. As such, it opens the possibility that the single date of 10,400 ± 150 BP (Gak-807) obtained on a charcoal sample from the base of Smith's 1965 sondage which corresponds to "an ashy zone at base of the mound" [34] is likely attributable to this lowermost unit of Ganj Dareh. This is reinforced by a review of the excavation records and excavation unit master list which describes Excavation Unit 1277 in square 15-H immediately to the western edge of the west trench depicted in our stratigraphy (Fig 4) as "soil & small stones bordering firepit = 1965 sondage" at a depth of ca. 7.40m below datum. This strongly suggests that the base of the 1965 sondage ('Ash 3'), where sample Gak-807 was recovered, correspond to the edge of the firepit that continues into 15-H from 15-I in our stratigraphy (Fig 14).

While this interpretation remains to be tested, it has the advantage of being coherent with the site's overall stratigraphy and Smith's field observations and it reconciles all of the dates from Ganj Dareh while providing a logical explanation for sample GaK-807, which after all must have dated *something*. If we are correct, this implies that a first phase of occupation at Ganj Dareh took place around 10,732–9807 cal. BC (95.4%), which would be somewhat older than the recently dated Early Neolithic 'boar pit' dated to 9660–9294 cal. BC (95.4%) at Asiab

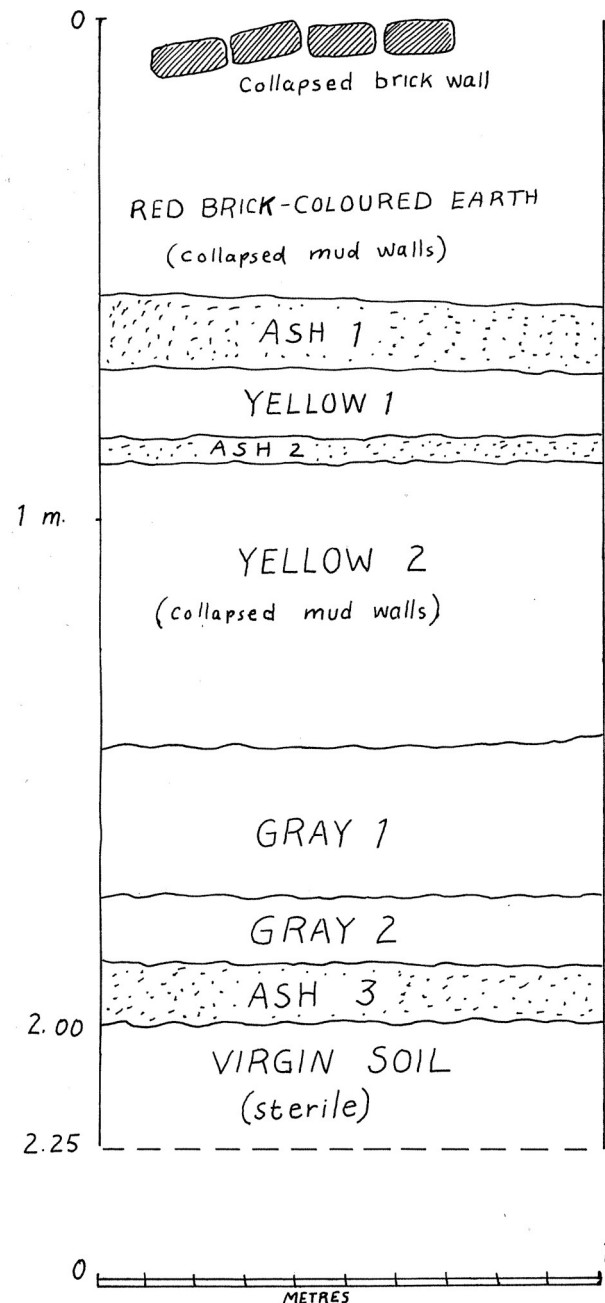

**Fig 14. Stratigraphic profile of the 1965 test pit in the 'west cut' (by P.E.L. Smith).**

[16] and the Early Neolithic deposits dated to 10,100–9140 cal. BC (95.4%) at Sheikh-e Abad [12]. At that time, Ganj Dareh would have likely been used by still mobile foragers as a seasonally task site, possibly as a pistachio nut processing station [48, 49]. This would conform with the 'high mobility' signature of the lithic assemblages recovered from the firepits, which is comparable only to that of the lithic assemblage from level P-01 (Fig 11). This would have been followed by a hiatus in occupation of some 2,000 years, after which the site was reoccupied by goat-herding pastoralists who, over the span of 200 years accumulated about 6.25m of deposit (or ca 3.01cm/yr). This scenario will be tested by ongoing analyses of the lithic

assemblages from Level H-01 to see whether they are distinct from those of Level E-01 and could be complemented by future zooarchaeological analyses contrasting the two levels.

## Burials

As summarized by Smith [31], the excavations at Ganj Dareh revealed 41 human primary and secondary burials, mostly found in Level D, some in flexed and some in extended positions. At least eight burials included more than one individual [80], and the inhumed individuals ranged in age from infants to adults, with several infant burials being deposited in subfloor cubicles [30]. Grave goods were very scant at Ganj Dareh, and exclusively found with non-adults: one child recovered in 1974 (Ganj Dareh 39) was found buried with an elaborate shell and stone bead necklace that included at least five *Oliva* shells [31], while one skeleton part of a triple burial interred in a kind of mudbrick 'sarcophagus' recovered in 1971 in Level C-01 was recovered with a polished stone pendant in its rib cage [30]. In addition, the burial of one tightly-flexed adult has been reported to have been deposited on reed matting or coarse textile [31]. A particularly noteworthy fact to emphasize is that at least 14 of the individuals buried at Ganj Dareh also show traces of artificial cranial deformation [78, 80]. Adding isolated elements and tabulating duplicate elements from burial contexts, Merrett [77] determined that a minimum of 116 human individuals were recovered during Smith's excavations at Ganj Dareh.

In our analysis of the West Trench, we identified and documented two burials discovered in 1971: a single inhumation in Level B-02 (individual 'Ganj Dareh 14') and a triple burial in Level C-01 (individuals 'Ganj Dareh 15, 16, and 17'). The triple burial corresponds to the 'sarcophagus' burial alluded to above. Our documentation allowed us to cross-reference information about the burial context, cranial deformation and bioarchaeological data, to create the first complete overview of the funerary context of these four individuals.

The burial recovered in Level B-02 was described in the field as belonging to a child and was recovered in a shallow depression in the northwest corner of square 15-I that extends into the northern section of the West Trench (Excavation Units 665 and 679). The individual (Ganj Dareh 14b, to use Merrett's designation [77]) is represented by a well-preserved cranium, the mandible, ribs and at least one lower long bone and was oriented with its head pointing south to and feet to the north (Fig 15). Bioarchaeological analyses converge on an age-at-death of approximately 3.5–4 years of age [77, 107]. Furthermore, Lambert [78] and Meiklejohn et al. [80] identified markers of artificial cranial deformation on GD 14b, specifically post-coronal depression, parietal bulging, lambdoid flattening and horizontal grooving on the pario-temporals. This has been interpreted as an unintentional consequence of habitual clothing or adornment in childhood or increasing community social identity rather than as a status marker [77, 80].

In contrast, the triple burial from Level C-01 was oriented northeast to southwest and contained one unsexed juvenile (Ganj Dareh 16; AOD: 6.5 years) and two teenage male individuals (GD 15 –AOD: 15 years; GD 17 –AOD: 13.5 years), one of whom (GD 17) was buried wearing a polished stone pendant that was recovered in his thoracic cavity (Fig 16). GD 15 was laid down on its right side facing north in an extended position, with its arms folded under his head, which was pointing east. In contrast, GD 16 was deposited in a flexed position, with its head pointing south, and GD 17's position and orientation were impossible to reconstruct because the remains were disturbed by the incorporation of the other two individuals in the burial. While it is hard to determine based on the field documentation alone, a likely order of interment appears to have been GD 17 being buried first, followed by GD 16 which disturbed its original position, while GD 15 was finally laid supine above the other two individuals. This sequence of burial also implies that the three individuals were not buried at exactly the same moment, but rather over some period of time. Like GD 14b, the crania of all three individuals

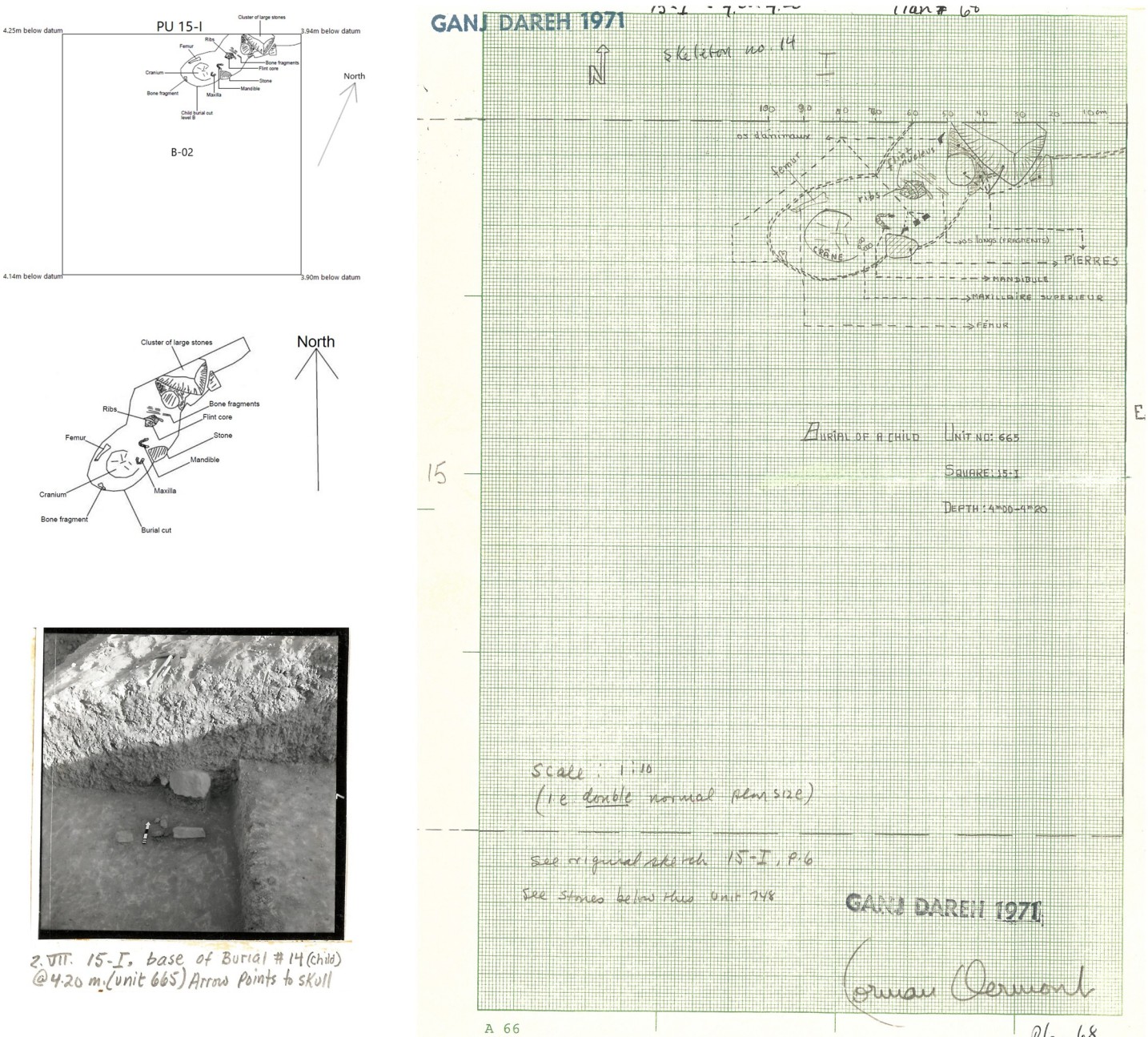

**Fig 15. Child burial in level B-02.** A. Position of burial in planimetric unit 15-I. B. Detail of burial. C. Archive photograph of burial. D. Field notes detailing burial.

interred in this burial show evidence of cranial deformation [80], while GD 15 also exhibits a well-healed fracture on its left humerus [107].

Beyond containing these three individuals, this burial is also unusual in that it included structural elements that were originally interpreted as a form of mudbrick "sarcophagus," which would be a unique occurrence at Ganj Dareh:

"In one case three extended skeletons (an adult, an adolescent and a child) were found together inside a curious elongated "sarcophagus" made of mud bricks and covered with a

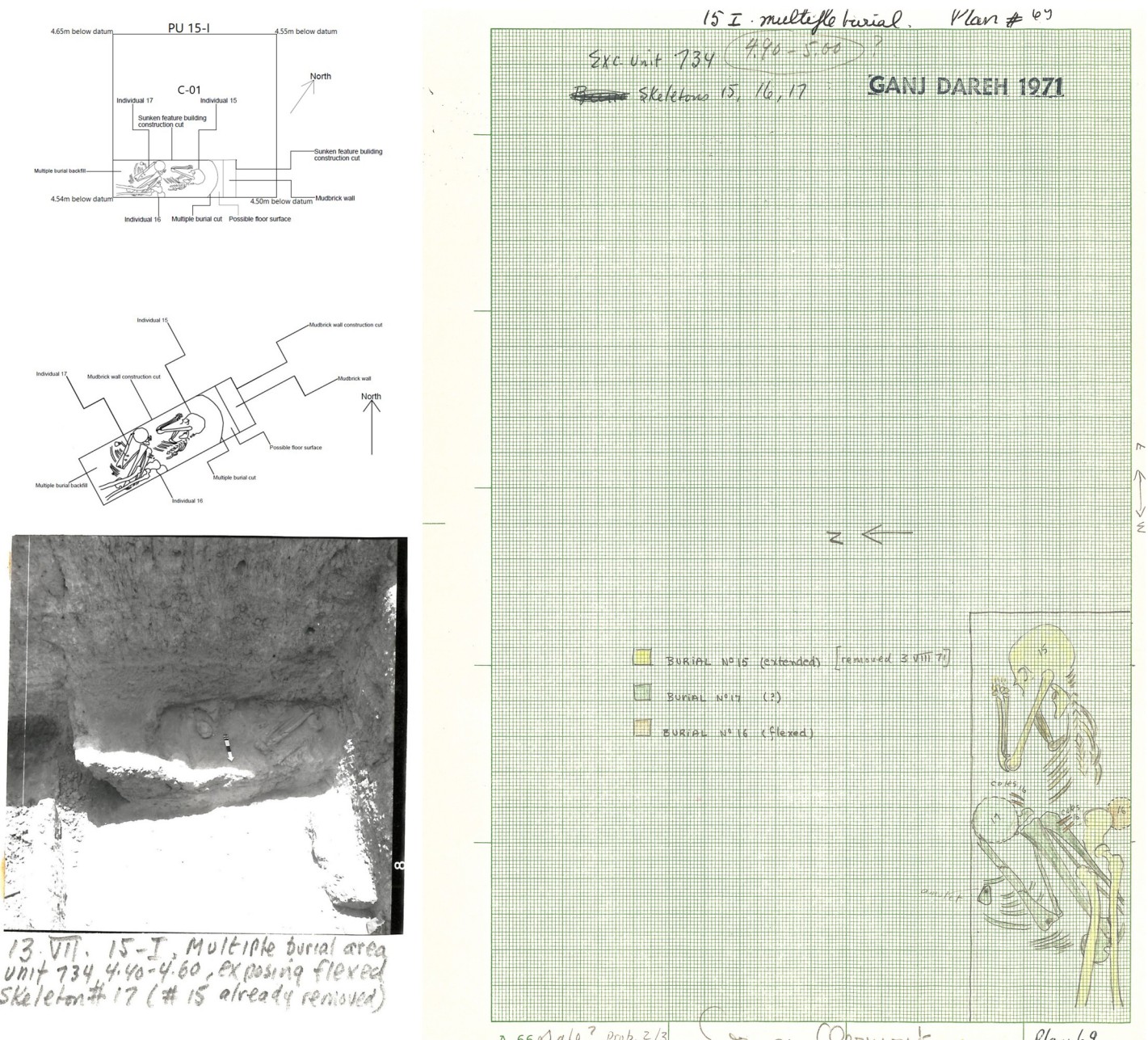

**Fig 16. Triple burial from level C-01.** A. Position of burial in planimetric unit 15-I. B. Detail of burial. C. Archive photograph of the burial. D. Field notes detailing burial.

kind of mud roof. The bodies rested on a thick layer of greyish powdery material, perhaps burned limestone, and one had a finely polished stone pendant in its rib cage, the only evidence of grave goods so far found at Ganj Dareh [30]."

Our analysis confirms that the triple burial shares a stratigraphic relationship with a mudbrick wall, which is what the "sarcophagus" claim is based on, by virtue of its proximity to the burial. However, a closer inspection of the stratigraphy indicates a different relationship between the triple burial and the mudbrick wall. Indeed, doing so reveals that a cut was first

dug for the mudbrick wall foundation. This cut was then filled by both the wall foundation itself and a reddish deposit, which may either be an altered buried floor surface, or the fill of the wall foundation cut (Fig 17A). This reddish deposit was then cut into by the multiple burial, prior to then being filled in, leaving a clear berm of sediments between the wall and the cut of the burial pit (Fig 17B). This stratigraphically-based sequence of events indicates that the wall and the burial were therefore not contemporaneous, with the wall having been built

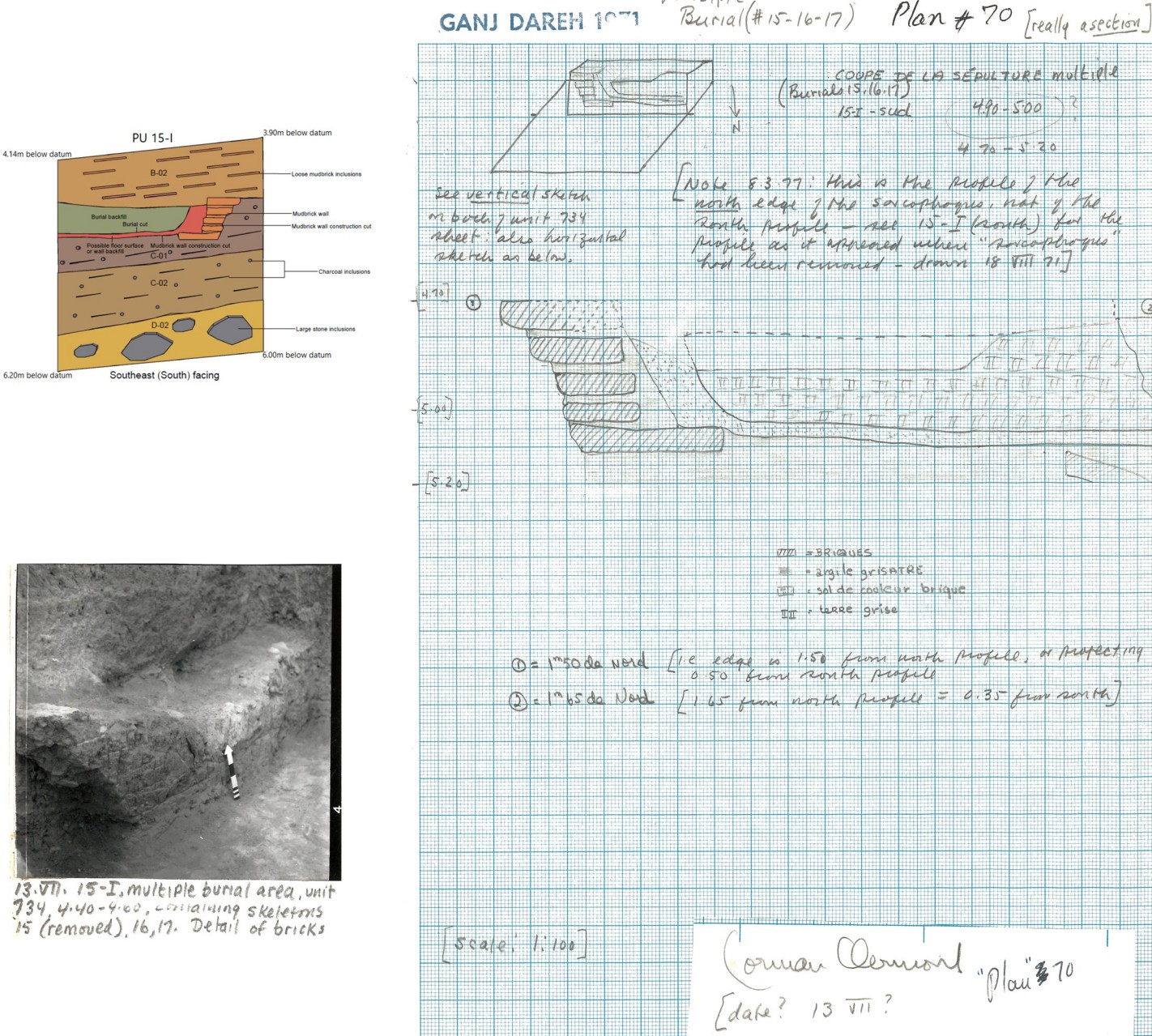

**Fig 17.** A. Level C-01 burial, composite section (viewed from the south). B. Field photograph of structure of the burial (looking from the north). C. Field notes detailing the 'sarcophagus'.

seemingly well before the excavation of the burial, which was cut into sediment that had accumulated within the confines of the room it helped define.

A growing corpus of evidence about Aceramic Neolithic burial practices in the Zagros allows us to situate the Ganj Dareh patterns into a broader context. For instance, the data for the triple burial in Level C-01 indicate it was more likely dug within the confines of a living space, which Merrett [77] suggests could have been an abandoned house or a special mortuary chamber. That feature's interpretation as a possible mortuary chamber could be bolstered by the recent identification of such a 'charnel place' at the Aceramic Neolithic site of Bestansur in Iraqi Kurdistan, where the remains of at least 48 individuals were found comingled [18]. At Ganj Dareh, however, this space would have been plastered over and sealed following its punctual use, and it is possible that this feature extended further south into square 14-I, which remains unexcavated (Fig 2). In any case, whether the triple burial was intentionally placed in relationship to the wall remains an open question, but its location indicates a purposeful use of a ready-made structure. This might be another manner of embodying multigenerational bonds that linked Ganj Dareh's Aceramic Neolithic inhabitants over time.

Tepe Abdul Hosein, located about 100km away from Ganj Dareh at an elevation of 1860m asl in the same watershed, has yielded a slightly younger (8205–7755 cal BC) but similar sequence to that found at Ganj Dareh [19, 20]. It has also yielded a dozen Neolithic skeletons, including two triple burials from contexts described as 'fire collapse' (individuals 13029, 13030 Sk. 1 and 13030 Sk.2) or 'mud brick tumble' (individuals 19001 Sk. 1, 19001 Sk. 2 and 19001 Sk.3), which may have represented family units though the contextual and biological data to affirm this are wanting [19, 21]. Significantly, five of the identified adult males at Abdul Hosein also show clear evidence for pronounced cranial deformation which would have been more visible than that documented at Ganj Dareh [21].

At East Chia Sabz, two burials were discovered in Trench II [103]: a poorly preserved and completely burnt individual located next to a hearth and a second individual (directly dated to 8485–8293 cal. BC) who was deposited on its right side in a flexed position and oriented NW-SE. This burial was incorporated into a concentration of pebbles (perhaps a stone platform) and buried with a necklace comprising of stone beads and perforated terrestrial gastropod shells [103]).

At Sheikh-e Abad, located only a few kilometers NW of Ganj Dareh, six burials dated to about 7600 cal BC have been recovered, five of which were found in small rooms in Building 1, including two that cut through parts of its walls [108]. The burial of an infant (Burial 801), in contrast, was recovered from a space below the floor of Space 15. Two of the burials are associated with traces of red ocher and another laid on a surface of black organic matter that may be the remains of matting [18]. Four of the burials were deposited on their sides in a flexed position with their legs folded to the chest and oriented either E or W. [108], while Burial 712 was buried with its head to the N and feet to the S. Burial 707 stands out in this context for having been buried flexed, but laid on its back with its torso supine and its cranium missing, which has been interpreted as evidence of intentional cranial removal, as in the Levantine Pre-Pottery Neolithic [18, 108].

In sum, the new information on burial practices at Ganj Dareh contribute to a better, more detailed understanding of the diversity of Aceramic Neolithic mortuary practices in the Zagros. Details about the triple burial from Level C-01 show it shares superficial similarities to some of the triple burials from Abdul Hosein [21], although the complexity of the structure encasing it suggests it may rather represents a kind of dedicated mortuary chamber, which suggests possible affinities to the 'charnel house' from Bestansur [18]. Regardless of this feature's original intended use, this study clearly demonstrates that the materials comprised in the Ganj Dareh archives at Université de Montréal can be used to shed important new light on the social life and cultural evolution at the site and the region as a whole.

## Conclusion

The results of this study show the considerable scientific potential of the Ganj Dareh archive at Université de Montréal to helping us refine our understanding of this site's human occupational history and, by extensions, of the emergence of the Neolithic in the Central Zagros. The available documentation allowed us to reconstruct the first detailed stratigraphy of Ganj Dareh's West Trench. This has shown that its stratigraphy is more complex than Smith's original five-level sequence, comprising as many as fourteen stratigraphic horizons. This reconstruction cross-tabulated with artifact density records has further enabled us to discern subtle changes in technological organization over time, notably a clear distinction between the more mobile occupants of Levels P-01 and H-01 and the more sedentary communities of the overlying layers (D-06 to B-01). In addition, clay technology appears to have emerged suddenly, with ceramic artifacts found in all levels overlying H-01.

Thus, a clear change in technological organization occurred after H-01, likely in response to the growing importance of sedentism from Level E-01 on. This shift in residential practices accompanied by the deposition of burials in living areas in Levels C-01 and B-02 suggest the latter may have been symbolic markers of permanence and place. These burials constitute the first detailed presentation of funerary practices at Ganj Dareh, allowing us to insert them into the growing diversity of Aceramic Neolithic mortuary rituals in the Zagros.

The West Trench stratigraphy cross-referenced with field documentation also allows us to propose a novel view on some of the discrepancies between the radiocarbon dates obtained for Ganj Dareh over the years, and in particular of the "aberrant" date of 10,400 ± 150 BP (Gak-807) taken from the firepit horizon at ca. -7.40m below datum. Our analysis suggests that the firepits may have been created by one or several highly mobile Epipaleolithic communities, that used them seasonally [48], while the overlying layers were deposited either by several generations of the same or a succession of Aceramic Neolithic communities that occupied the site a few centuries later. This resurrects the idea that there may in fact have been a hiatus in occupation at the site between Level H-01 and E-01, a working hypothesis which future field research at the site should be able to test in relatively short order (e.g., [15]). Finally, this fine-grained understanding of Ganj Dareh's stratigraphy has enabled us reconstruct a detailed sequence of occupation levels and explore the horizontal variability of each, allowing us to posit the stratigraphic relationship between a single mudbrick wall, its construction cut backfill and a successive multiple burial that was subsequently plastered over.

Much work remains to be done on these collections, but the present analysis now serves as an anchor for future studies of Ganj Dareh's material cultural and site formation history. Going forward, it opens up several productive avenues of research. For instance, with the collected stratigraphic data, it now becomes possible to conduct a stratigraphically informed Bayesian analysis of a large corpus of radiocarbon dates from the site. This could help constrain the length of occupation of different levels in spite of the apparent overlap between the radiocarbon dates taken from the levels above H-01 [38]. This would enable a better understanding of the duration of specific phases of Aceramic Neolithic occupations at the site. Moreover, a reanalysis of the palynological record in relation to the reconstructed stratigraphy could be conducted to discern whether any tangible vegetational changes occurred after the emergence of mudbrick architecture and goat herding, with attendant implications for some recent scenarios about human niche construction around the site [49, 50].

It should be emphasized that this paper is part of only the initial phase of study of the material included in the Ganj Dareh collections and archive stored at Université de Montréal, under the supervision of the lead author. The collections include tens of thousands of chipped stone tools (including sickle blades, formal tools, débitage and cores) and of ground stone

tools (pestles, etc.), in addition to hundreds of clay tokens, sherds, and human and animal figu-rines, as well as bone tools and ornaments of various sorts. The human remains are curated in the Simon Fraser University Department of Archaeology (Burnaby, BC, Canada), while most of the unworked faunal remains are currently curated at the Smithsonian Institution (Wash-ington, DC, USA). The collection housed at Université de Montréal also comprises dozens of sedimentary, architectural and charcoal samples, while the archives comprise the complete field notes, unit records, hand-drawn plans and sections, photographs, correspondence, and photographic slides, among other documentation pertaining to work at the site from between 1965 and 1974. A preliminary inventory indicates that some photographs appear to be missing from the archive, however. The analysis presented in this paper based on this archival material also underscores the importance for archaeological projects to conserve hard copies of field documentation post-excavation to permit the analysis of archaeological material by future researchers should a site not be fully published in a timely manner [109–111]. Given the grow-ing recognition that archaeological legacy collections offer the potential to help address con-temporary issues [112], it is essential to ensure their 'legibility' for future archaeologists by associating documentation in a format that is not strictly dependent on proprietary software.

As indicated above, the West Trench that was the focus of this first study represents less than 10% of the entire excavated area of Ganj Dareh. This underscores that there still remains a wealth of information to be gleaned from the in-depth study of both the artifacts and the doc-umentation that will help clarify both the occupational history of Ganj Dareh and the Neolithi-zation process in the Zagros more broadly. The quality of the data to be extracted from the Ganj Dareh archive is further highlighted by the fact that two PhD and several MSc projects by Université de Montréal students based on them are ongoing. The next phase of work based on the Ganj Dareh archive will focus on fleshing out our understanding of the diversity of mortu-ary practices at the site, as well as on testing the reality that Levels H-01 and E-01 represent dis-tinct, perhaps Epipaleolithic, kinds of occupations based on material drawn from entire area explored at the site.

## Supporting information

**S1 File. Data and code.**
(ZIP)

**S1 Video. Animated 3D model of the Smith's excavated levels at Ganj Dareh.**
(AVI)

## Acknowledgments

We thank Lyne Grondin for her priceless work digitizing the index card data and discussions about artifact counts, and Chris Meiklejohn, Deb Merrett, Tobias Richter, Hojjat Darabi and Ariane Burke for useful discussions about Ganj Dareh. We also thank Mehmet Özdoğan and Melinda Zeder for giving very useful feedback on the original manuscript.

## Author Contributions

**Conceptualization:** Julien Riel-Salvatore.

**Data curation:** Julien Riel-Salvatore, Andrew Lythe, Alejandra Uribe Albornoz.

**Formal analysis:** Julien Riel-Salvatore, Andrew Lythe, Alejandra Uribe Albornoz.

**Funding acquisition:** Julien Riel-Salvatore.

**Investigation:** Julien Riel-Salvatore, Andrew Lythe, Alejandra Uribe Albornoz.

**Methodology:** Julien Riel-Salvatore, Alejandra Uribe Albornoz.

**Project administration:** Julien Riel-Salvatore.

**Software:** Alejandra Uribe Albornoz.

**Supervision:** Julien Riel-Salvatore.

**Validation:** Julien Riel-Salvatore, Andrew Lythe, Alejandra Uribe Albornoz.

**Visualization:** Andrew Lythe, Alejandra Uribe Albornoz.

**Writing – original draft:** Julien Riel-Salvatore, Andrew Lythe.

**Writing – review & editing:** Julien Riel-Salvatore, Andrew Lythe, Alejandra Uribe Albornoz.

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
