## [Decision Letter · Decision Letter 0]

1 Oct 2020

PONE-D-20-23322

New Insights into the Pre-Pottery Neolithic Spatial Organization, Stratigraphy and Human Occupations of Ganj Dareh, Iran

PLOS ONE

Dear Dr. Riel-Salvatore,

Thank you for submitting your manuscript to PLOS ONE. After careful consideration, we feel that it has merit but does not fully meet PLOS ONE’s publication criteria as it currently stands. Therefore, we invite you to submit a revised version of the manuscript that addresses the points raised during the review process.

All comments need to be addressed before re-submission.

We look forward to receiving your revised manuscript.

Kind regards,

Peter F. Biehl, PhD

Academic Editor

PLOS ONE

Additional Editor Comments:

Your manuscript has now been seen by two referees, whose comments are appended below. You will see from these comments that the referees find your work of great interest, but have raised several issues that must be addressed before publication.

Journal Requirements:

"This work was funded and supported by Université de Montréal, and the Canadian

Foundation for Innovation JELF grant 37754 (to JRS). AUA is supported by a Joseph-

Armand Bombardier Canada Graduate Scholarship."

"JRS: Canadian Foundation for Innovation JELF grant 37754. https://www.innovation.ca/awards/john-r-evans-leaders-fund

The funder had no role in study design, data collection and analysis, decision to publish, or preparation of the manuscript."

5. Please upload a new copy of Figure 2b as the detail is not clear. Please follow the link for more information: https://blogs.plos.org/plos/2019/06/looking-good-tips-for-creating-your-plos-figures-graphics.

6. We note that Figure 2a in your submission contain satellite images which may be copyrighted. All PLOS content is published under the Creative Commons Attribution License (CC BY 4.0), which means that the manuscript, images, and Supporting Information files will be freely available online, and any third party is permitted to access, download, copy, distribute, and use these materials in any way, even commercially, with proper attribution. For these reasons, we cannot publish previously copyrighted maps or satellite images created using proprietary data, such as Google software (Google Maps, Street View, and Earth). For more information, see our copyright guidelines: http://journals.plos.org/plosone/s/licenses-and-copyright.

6.1.    You may seek permission from the original copyright holder of Figure 2a to publish the content specifically under the CC BY 4.0 license. 

6.2.    If you are unable to obtain permission from the original copyright holder to publish these figures under the CC BY 4.0 license or if the copyright holder’s requirements are incompatible with the CC BY 4.0 license, please either i) remove the figure or ii) supply a replacement figure that complies with the CC BY 4.0 license. Please check copyright information on all replacement figures and update the figure caption with source information. If applicable, please specify in the figure caption text when a figure is similar but not identical to the original image and is therefore for illustrative purposes only.

Reviewers' comments:

Reviewer's Responses to Questions

**Comments to the Author**

1. Is the manuscript technically sound, and do the data support the conclusions?

Reviewer #1: Yes

Reviewer #2: Yes

2. Has the statistical analysis been performed appropriately and rigorously? 

Reviewer #1: N/A

Reviewer #2: Yes

3. Have the authors made all data underlying the findings in their manuscript fully available?

Reviewer #1: No

Reviewer #2: Yes

4. Is the manuscript presented in an intelligible fashion and written in standard English?

Reviewer #1: Yes

Reviewer #2: Yes

5. Review Comments to the Author

Reviewer #1: Review of Riel-Salvatore et al.

This initial report on the analysis of the newly discovered excavation reports from the famous site of Ganj Dareh is a very welcome addition to the growing number of studies of the Neolithic in Central Zagros. The Central Zagros was once held to be a major center for initial domestication and agricultural emergence in the Near East. With the interruption of archaeological research in the region in the late 1970s the focus of research shifted to the Levant and the Central Zagros, and indeed the entire eastern half of the Fertile Crescent region, was portrayed as a backwater, bit-player in the story of the Near Eastern Neolithic. Thanks to a combination of new investigations in the region and the reanalysis of material from earlier excavations (like this one), we are seeing a major rehabilitation of the Central Zagros as a major player in the multi-centric process of domestication and agricultural emergence that took place across the entire arc of the Fertile Crescent during the Early Holocene.

Ganj Dareh has long been recognized as an important site for understanding the Iranian Neolithic. Yet other than the analysis of the faunal remains by Hesse and then Zeder, there has been almost no detailed published account of the stratigraphy and material culture recovered in early excavations at this site. This careful analysis of the excavation reports recently discovered at the University of Montreal is then a significant contribution to our growing understanding of the Iranian Neolithic and its role in the emergence of agricultural economies and village life in the Near East. This report fills in major gaps in our understanding of Ganj Dareh and its place in the process of Neolithization in the Near East by providing at long last a report on the stratigraphy of this complicated tell site, as well as information on the lithics, clay objects, chronology, and burials gleaned from these records. Moreover, the discovery of the two previously unrecognized lower levels and their possible affiliation with a preceding Epipaleolithic period that may predate the upper architectural levels of the site may at long last resolve a long open question about an apparently anomalous early date from basal levels and, more importantly, help brings into sharper focus the picture of the foundational Early Holocene adaptations in the Zagros that in all likelihood saw the initial steps toward both plant and animal domestication in this region. As such, I fully support publication of this report in PLOS ONE.

I do, however, have a few suggestions for revision.

My major recommendation is that more be done to set this study within the broader context of the Neolithic in the Central Zagros. The picture of agricultural emergence in the region and Ganj Dareh’s place within it presented here seems somewhat dated and does not make effective use of recent work in the region. In their discussion of the archaeobotanical remains from the site, for example, the authors downplay the role of domesticated crops at Ganj Dareh maintaining that the only evidence of food production at the site is the presence of domesticated goats. To support this thesis they cite van Zeist’s hypothesis that the domestic barley that increases in prevalence throughout the occupation of the site was likely not for human consumption but used as animal fodder. And yet there is now increasing evidence for the active engagement of people in the region in low level food production that involves native grasses and domestic barely documented at Chogha Golan and other recently excavated sites in the region. And while the barley recovered from dung burned as fuel at Ganj Dareh (and other contemporary sites) was likely to have been fed to goats, that doesn’t mean that humans didn’t consume barley as well.

They claim that the lentils recovered from the site were likely wild collected resources on the basis of their small size. This ignores important work by Weiss in the Levant that makes the case for early cultivation/domestication of lentils despite their small size research that indicates that an increase in seed size is a lagging indicator of domestication in this species. They also fail to cite important work by Savard and others that shows that people in this region had a heavy focus on pulses (including lentils) reaching back into the Epipaleolithic indicative of the management if not indigenous domestication of these plants.

Moreover, while the faunal remains from Ganj Dareh do indeed show the earliest clear evidence of goat management in the harvest profiles congruent with strategies of modern day herders, it is unlikely that the site captures the initial phases of goat domestication as implied here. The fact that this harvest strategy is fully in-place in the earliest levels at the site suggest that this practice considerably pre-dates the architectural levels at the site. This possibility is reinforced by data from southeastern Anatolia that indicates the presence of managed goats by 10,500 cal. BP and for the importation of clearly managed goats to Cyprus around the same time.

The discussion of the environmental setting of the site fails to make reference to recent paleo-environmental reconstructions by Roberts and Asouti that have considerably revised that of Van Zeist in the cited 1984 publication.

Similarly, more discussion is needed of the importance of the exciting discovery of the previously unrecognized basal levels of the site. The identification of these levels (along with the intriguing lithic evidence for greater mobility than in the later levels E though A) fits well into the emerging picture of the Epipaleolithic in the region documented by more recent excavations at Asiab, basal Sheikh-e-Abad, Chia Sabz, and Chogha Golan, as well as by older excavations at Abdul Hosein. At all these sites we find evidence for an early phase of the settlement history of the region in which foragers made use of the increasing abundance of resources that accompanied Early Holocene climate amelioration. These discoveries suggest that the Central Zagros is likely part of a broader cultural phenomenon spread across the Eastern Fertile Crescent region – Peasnall’s round house phase – evidenced at contemporary sites such as Zawi Chemi, Karim Shahir, Hallan Çemi and even basal levels at Çayönü. These are the sites (and the time period) that likely served as the starting point for the food production strategies that are firmly in place by the time of the establishment of the architectural levels at Ganj Dareh.

The picture that emerges, then, is that these newly discovered levels at the site represent this initial phase of Early Holocene settlement history in the region, while the architectural levels represent a later phase in which low level food production that feature a mix of domestic, managed, and wild resources is well established among a network of fully sedentary villages spread across the Central Zagros. More discussion of this broader context would help underscore the importance of this analysis of these archival materials in adding to the growing understanding of the Neolithic in the Central Zagros.

On a less substantive level, it would be good to have a clearer statement whether this paper represents just an initial phase of the study of the materials discovered in Montreal – that of the excavation records and not of the artifacts that accompany these records. Are these all the field excavation reports, plans, photos, etc. or just a selection of them? It would also be good to have an account of what the archival materials consist of as well as a discussion of any plans for future study of these archives. Some idea of the size of the artifact collections in Canada and plans for future work on them would also be helpful.

The paper would also benefit from more illustrations of the lithics (especially the purported Paleolithic stone tools) and the clay objects discussed in the text.

Reviewer #2: I have found the paper to be of high significance, firstly because the site of Ganj Dareh had featured as a complex entity captivating considerable interest since several decades on a number of issues, and also because there was some ambiguity in reading the original publications. At the time when the site was excavated, our knowledge on the cultural horizon of PPN was still in infancy, though arousing considerable excitement. The site, along with its stand on early goat domestication, had stimulated a rather controversial discussion, together with the sister site of Tepe Sarab, on the early use of ceramic objects, mainly clay tokens. As noted in the manuscript, soon after the termination of excavations, archaeological research had been interrupted in the region for some decades, thus our understanding and assessments being left to depend on the level of 1970's. Accordingly, this manuscript has been most welcome in bringing back the site into agenda and in reassessing the original field documentation through a new perspective. Likewise presenting a critical overview on what had been said through time on various artefact categories, and to consider them by correlating with original field documentation has been a very significant and stimulating contribution to our understanding of this critical site.

The paper has been well organized, easy to follow, though presenting substantial data, has successfully avoided to be lost in details. In this respect, some of the appended simulations, though presenting a colourful nice picture, actually do not contribute much to our understanding of either the site or of its stratigraphy.

Another issue that needs to be acknowledges is the importance of keeping original field documentation. Excavating a site, evidently implies erasing or removing remains what had been preserved through time; and regardless of our concerns to be objective, we are bound by the trends of our times. The present manuscript thus presents anew view almost half a century after its original publication, but still being based on original documentation. This deserves to be high acknowledged, also hoping to be an example for other old excavations including those off mine.

In page 30, in the section concerning lithics, using the " lithic density per cubic meter of excavated sediment" analysis to interpret mobility or sedetary way ıof living, I have found to be highly subjective, as the desit of any artefact category would vary considerably on functionality of that articular space and not necessarily on social structuring of the society.

6. PLOS authors have the option to publish the peer review history of their article (what does this mean?). If published, this will include your full peer review and any attached files.

Reviewer #1: **Yes: **Melinda Zeder

Reviewer #2: **Yes: **Emeritus Prof. Dr. Mehmet Özdoğan

---

## [Author Response · Author response to Decision Letter 0]

22 Apr 2021

Dear Professor Biehl,

We write the following rebuttal letter to respond to the comments on our manuscript “New Insights into the Pre-Pottery Neolithic Spatial Organization, Stratigraphy and Human Occupations of Ganj Dareh, Iran” which you transmitted to us on Oct. 1, 2020, indicating that addressing them involved making minor revisions to the manuscript. We begin by thanking the reviewers for their extremely positive assessments and for their constructive feedback. We also would like to apologize for the time it took us to return the revised manuscript, which was in part due to two of the co-authors coming down with COVID-19 and tracking long-term symptoms since. 

Prof. Özdoğan’s comments were minimal and straightforward to address. Concerning the animated 3D reconstruction of the site’s stratigraphy, we have added on p. 17 a comment about the detail this animation brings to our understanding of the site’s overall layout and changing occupation focus, which is more difficult to appreciate using static plans. On p. 47, we have also added a paragraph explaining the importance of conserving analog hard copies of field documentations, in accordance with best practices in legacy collection management. We also detail how this fits into a broader plan of future work for the Ganj Dareh archive at UdeM in an additional two concluding paragraphs on pp. 46-47 detailing the next steps of this work over the coming years, in response to Dr. Zeder’s comments. Lastly, concerning the issue of lithic density being influenced by area function, we have already cited on p.31 the relevant literature demonstrating the validity of this approach in a variety of archaeological contexts, but we have added a qualifying sentence that highlights that any lithic assemblage is liable to be biased thusly, whether it is analyzed using the WABI as we do here, or using more traditional techno-typological approaches. 

Turning to Dr. Zeder’s very helpful comments and suggestions, we have updated the relevant sections of the text and incorporated her suggested literature on research on the Aceramic Neolithic of the Zagros and the Eastern Fertile Crescent. On pp. 7-8, we have updated the discussion about the likely importance of domesticated crops at Ganj Dareh and other nearby Aceramic Neolithic sites, focusing on pulses as well as recently published data on triticoid grains. This is complemented by a broader discussion of the context for goat domestication in Anatolia and neighboring region at that time to better situate the evidence for this phenomenon at Ganj Dareh on p. 6. Additional context has also been added by reference to recent paleoenvironmental research on the Aceramic Neolithic of the Zagros on p. 3. In cultural terms, we have added a section on p. 6 that explains how Ganj Dareh was the full expression of a cultural phenomenon that unfolded across the Eastern Fertile Crescent in the preceding centuries. We have, in particular, expanded the discussion on pp. 33, about how the pattern of possible Epipaleolithic assemblages at the base of Ganj Dareh corresponds with has been documented at sites in the region. 

As concerns the illustrations of the lithics and clay objects, these are part of ongoing unpublished efforts (MSc and PhD theses) and we prefer to reserve them for publication where the students who produced them can be first author, especially since they are simply the subject of basic quantification here, not detailed analysis. We felt comfortable with this decision since Prof. Özdoğan did not mention this as an essential revision. We have also taken the liberty to add a few references to new studies pertaining to the archaeology of Ganj Dareh and the Central Zagros that have been published since we submitted our manuscript this past summer. 

Turning to your editorial comments, we have modified Fig. 2a in accordance to your specification and removed its complement originally submitted as Fig. 2b. As such, we have relabeled that revised figure, Fig. 2 (and modified referring text on p. 16). We have also modified the acknowledgments on p. 48 to remove mentions to funding sources and to thank the reviewers and two other colleagues who have provided help in contextualizing the site. On p. 11, we have added a sentence indicating we are studying the Ganj Dareh material with the permission of the Faculty of Arts and Sciences at Université de Montréal.

All of the modified text has been highlighted in red, with removals highlighted in red and double-crossed through. All of the numbering of the references has been changed in order to account for the addition of over 25 new sources.

We hope you will find what have addressed your and the reviewers’ comments satisfactorily and look forward to hearing about your final decision about publishing our manuscript in PLOS ONE in the near future. 

On behalf of my co-authors and I,

Bien cordialement,

Julien Riel-Salvatore, PhD

Professeur titulaire

---

## [Editor Report · Decision Letter 1]

26 Apr 2021

New Insights into the Spatial Organization, Stratigraphy and Human Occupations of the Aceramic Neolithic at Ganj Dareh, Iran

PONE-D-20-23322R1

Dear Dr. Riel-Salvatore,

We’re pleased to inform you that your manuscript has been judged scientifically suitable for publication and will be formally accepted for publication once it meets all outstanding technical requirements.

Kind regards,

Peter F. Biehl, PhD

Academic Editor

PLOS ONE
---

## [Editor Report · Acceptance letter]

28 Jul 2021

PONE-D-20-23322R1 

New Insights into the Spatial Organization, Stratigraphy and Human Occupations of the Aceramic Neolithic at Ganj Dareh, Iran 

Dear Dr. Riel-Salvatore:

I'm pleased to inform you that your manuscript has been deemed suitable for publication in PLOS ONE. Congratulations! Your manuscript is now with our production department. 

Kind regards, 

on behalf of

Dr. Peter F. Biehl 

Academic Editor

PLOS ONE